# Human pre-valvular endocardial cells derived from pluripotent stem cells recapitulate cardiac pathophysiological valvulogenesis

Tui Neri[1,2,13], Emilye Hiriart[1,13], Patrick P. van Vliet[3,4,5,6], Emilie Faure[1], Russell A. Norris[7], Batoul Farhat[1,5,6], Bernd Jagla[8], Julie Lefrancois[1], Yukiko Sugi[7], Thomas Moore-Morris[1,5,6], Stéphane Zaffran [1], Randolph S. Faustino[9], Alexander C. Zambon[10], Jean-Pierre Desvignes[1], David Salgado [1], Robert A. Levine[11], Jose Luis de la Pompa [12], André Terzic[9], Sylvia M. Evans[3], Roger Markwald[7] & Michel Pucéat [1,5,6]

Genetically modified mice have advanced our understanding of valve development and disease. Yet, human pathophysiological valvulogenesis remains poorly understood. Here we report that, by combining single cell sequencing and in vivo approaches, a population of human pre-valvular endocardial cells (HPVCs) can be derived from pluripotent stem cells. HPVCs express gene patterns conforming to the E9.0 mouse atrio-ventricular canal (AVC) endocardium signature. HPVCs treated with BMP2, cultured on mouse AVC cushions, or transplanted into the AVC of embryonic mouse hearts, undergo endothelial-to-mesenchymal transition and express markers of valve interstitial cells of different valvular layers, demonstrating cell specificity. Extending this model to patient-specific induced pluripotent stem cells recapitulates features of mitral valve prolapse and identified dysregulation of the SHH pathway. Concurrently increased ECM secretion can be rescued by SHH inhibition, thus providing a putative therapeutic target. In summary, we report a human cell model of valvulogenesis that faithfully recapitulates valve disease in a dish.

[1] INSERM U-1251, MMG, Aix-Marseille University, Marseille 13885, France. [2] Istituto di Ricerca Genetica e Biomedica, UOS di Milano, CNR, Rozzano 20138, Italy. [3] University of California San Diego, Skaggs School of Pharmacy and Pharmaceutical Sciences, La Jolla, CA 92092 92093, USA. [4] Cardiovascular Genetics, Department of Pediatrics, CHU Sainte-Justine, Montreal H7G 4W7 QC, Canada. [5] LIA (International Associated Laboratory) INSERM, Marseille U1251-13885, France. [6] LIA (International Associated Laboratory) Ste Justine Hospital, Montreal H7G 4W7, Canada. [7] Department of Anatomy and Cell Biology, Medical University of South Carolina, Charleston, SC 29401-5703, USA. [8] Institut Pasteur - Cytometry and Biomarkers Unit of Technology and Service, Center for Translational Science and Bioinformatics and Biostatistics Hub – C3BI, USR, 3756 IP CNRS, 75015 Paris, France. [9] Center for Regenerative Medicine, Mayo Clinic, Rochester, MN 55901, USA. [10] Department of Biopharmaceutical Sciences, Keck Graduate Institute, Claremont, CA 91711, USA. [11] Cardiac Ultrasound Laboratory, Harvard Medical School, Massachusetts General Hospital, Boston, MA 02111, USA. [12] Intercellular Signaling in Cardiovascular Development & Disease Laboratory, Centro Nacional de Investigaciones Cardiovasculares Carlos III (CNIC), Madrid E-28029, Spain. [13]These authors contributed equally: Tui Neri, Emilye Hiriart. Correspondence and requests for materials should be addressed to M.P. (email: michel. puceat@inserm.fr)

Congenital heart diseases are major causes of global mortality in children including in Europe and the United States[1,2]. Cardiac valves are affected in up to one third of these life-threatening conditions[3]. Moreover, in adults, the prevalence of valvular diseases dramatically increases with age reaching 13% of the elderly at 75 years of age or older[4,5]. Proper functioning of the valves is essential to ensure efficient blood pumping by the heart. The atrio-ventricular (AV) valves in humans feature two to three leaflets, which regulate the direction of blood flow through the mitral and tricuspid sides of the left and right ventricles, respectively. The semilunar valves direct the flow from the right (pulmonary valve) or the left (aortic valve) chamber. Despite significant progress in the field, there is still limited understanding of valve disease pathobiology.

Mitral valve prolapse (MVP) is among the most common conditions and affects 1 in 40 individuals[6]. Syndromic MVPs are observed in rare and genetically well-characterized connective tissue syndromes such as Marfan syndrome, Loeys–Dietz syndrome, or Ehlers–Danlos syndrome. Common non-syndromic MVP is a progressive disease originating from a compromised development of embryonic valves although functional consequences are apparent in middle-aged patients. Genetics studies have identified mutations in selected genes (*Filamin A*[7], *LMCD1*, *tensin*[8] and *Dachsous*[9]) with the disease manifesting by dysregulation of extracellular matrix (ECM) proteins in valve leaflets that precipitates myxomatous degeneration or fibroelastic deficiency. Ultimately, valve leakage is the cause of mitral regurgitation[10]. This requires surgical valve repair or replacement at an advanced stage of the disease as no preventive or curative pharmacological alternative is available.

Valve formation is a complex process that begins at embryonic stages HH16 in the chicken, E9.5 in the mouse, and at about 30 days in the human fetus. Intercellular signaling events occurring in the atrioventricular canal (AVC) and the outflow tract (OFT) initiate endocardial-to-mesenchymal transition (endoMT), whereby endocardial cells delaminate and invest the forming cushions. Later in cardiogenesis, this process encompasses interactions between different mesenchymal cell populations, morphogenesis, fusion of cushions, and ECM secretion, leading to the formation of mature and fully functional valves[11].

The endocardium, as the origin of valvular tissue, is formed by endothelial cells that are distinct from the traditional Flk1+ hemoangioblasts[12]. Two paradigms have been proposed to account for segregation of endocardial and myocardial lineages. First, studies in avian embryos[13–15] suggest that endocardial cells originate in the anterior lateral mesoderm. In line with this concept, *Nkx2.5* deletion in the mouse disrupts endocardial cushions[16]. Second, both endocardial and myocardial cells might share a common multipotent progenitor in the cardiac crescent. *Isl1Cre*-labeled cells as well as the *Mef2c-(AHF)Cre*-labeled counterparts give rise to both endocardial and myocardial cells[17–19], thus suggesting that the endocardium also originates at least partially from the second or anterior heart field (AHF) associated with the formation of the right ventricle and OFT, albeit that an exclusive AHF origin has been challenged. Instead, a contribution of endothelial cells derived from MesP1+ cells to the formation of the endocardium was proposed[19].

Genetically modified mouse models have provided important clues as to the cell lineages and signaling pathways that contribute to valve formation[20–22]. However, these murine transgenic models are limited in their potential to reveal mechanisms underlying complex processes, as aspects of signaling interaction, cell metabolism, epigenetics, and mechano-transduction are difficult to mechanistically separate in vivo and might not be identical in human valve development. In fact, cell lineages that contribute to the valves in humans have been elusive and have only been inferred from knockout mouse models recapitulating part of the human pathophenotype. A human specific cell model of valvulogenesis that could be extended to cells derived from individual patients would definitively advance the understanding of developmental mechanisms driving valvulogenesis in health and disease.

Pluripotent stem cells have been reported to recapitulate early developmental processes including cardiac myogenesis[22] but differentiation of these pluripotent cells towards valvular specific cells has not yet been reported. Here, leveraging embryology data across species, we report that human pluripotent stem cells are able to recapitulate the developmental process of valvulogenesis. We use a population of human pluripotent stem cell-derived MESP1+ sorted cardiovascular progenitors[23] and direct their fate concomitantly towards the first/second heart field and endothelial cell lineages. We report that these cells collectively or at the single cell level express a set of salient genes that mark pre-EMT (E9.0) mouse AVC. These cells undergo EMT in both notch-dependent and independent manners and express specific valve proteins when treated with BMP2, when seeded onto collagen hydrogels, or when grafted in vivo in mouse embryos. Furthermore, using patient-specific induced pluripotent stem (iPS) cells harboring a mutation in *Dachsous*[6] and in turn propensity to mitral valve prolapse[9], we recapitulate cell features of the valvulopathy, and identify a molecular mechanism of the disease that points to a therapeutic target. Human pluripotent stem cell (hPSC)-derived endocardial cells thus represent a representative model for human valvulogenesis enabling future studies on mechanisms in human valve pathogenesis.

## Results

**hPSCs differentiate into endocardial cells**. We developed here a protocol to differentiate human ES (hES) or induced pluripotent stem (hiPS) cells into genuine valvular cells. Undifferentiated pluripotent stem cells were first differentiated into MesP1+ cardiovascular progenitors using Wnt3a (100 ng/ml) the first day, then Wnt3a and BMP2 (10 ng/ml) and then BMP2 alone the third day; MESP1+ cells[23] were then sorted using the BMP2-induced SSEA-1 cell membrane antigen and plated on mouse embryonic fibroblasts (MEF) in fibronectin-coated plates. To segregate myocardial and endocardial cell lineages from the SSEA1+ MESP1+ cell population, cells were treated with VEGF (30 ng/ml), an inducer of endothelial cell fate. We further added FGF2 (2 ng/ml) and FGF8 (10 ng/ml) for 6 days. FGF8 was used to trigger mesodermal cell fate toward endocardial cells at the expense of myocardial cells[24]. Cells were then phenotyped using real-time RT-PCR, immunofluorescence and high content imaging. Figure 1a shows that VEGF/FGF2/FGF8-treated cells sorted with anti-CD31-conjugated beads intensely expressed *TBX2*, *TBX20, GATA5, MSX1, SMAD6, PITX2, HES1, NFATc, ADORA and GALANIN*, when compared to the level of expression of these genes in the SSEA1+-sorted MESP1+ cell population. A global transcriptomic analysis of the human VEGF/FGF2/FGF8-treated and CD31-sorted endocardial cells (named Human PreValvular Cells or HPVCs) versus MESP1+ cells revealed enrichment of *TGFβ1, TGFβ2, MSX1, THROMBOSPONDIN, PITX2, ERBB4, ADAM19*, and *CAV1* in HPVCs, similar to the endocardial expression signature in the mouse AVC endocardium[25,26]. *PDGFRα, LMCD1* and its target *GATA6*, were enriched in cardiac cushions at the level of mRNA[27] as were endothelial genes *KDR, FLT1, FLI1, TIE2, NFATc* (Fig. 1b; Supplementary Fig. 3, transcriptomic data GEO dataset). Conversely, *MESP1*, and cardiac chamber specific genes *TBX5, MEF2c, NPPA, MLC2v* were not expressed in HPVCs (see transcriptomes). Comparisons of HPVC transcriptomes with H1 and H9 ESC-derived mesenchymal cells

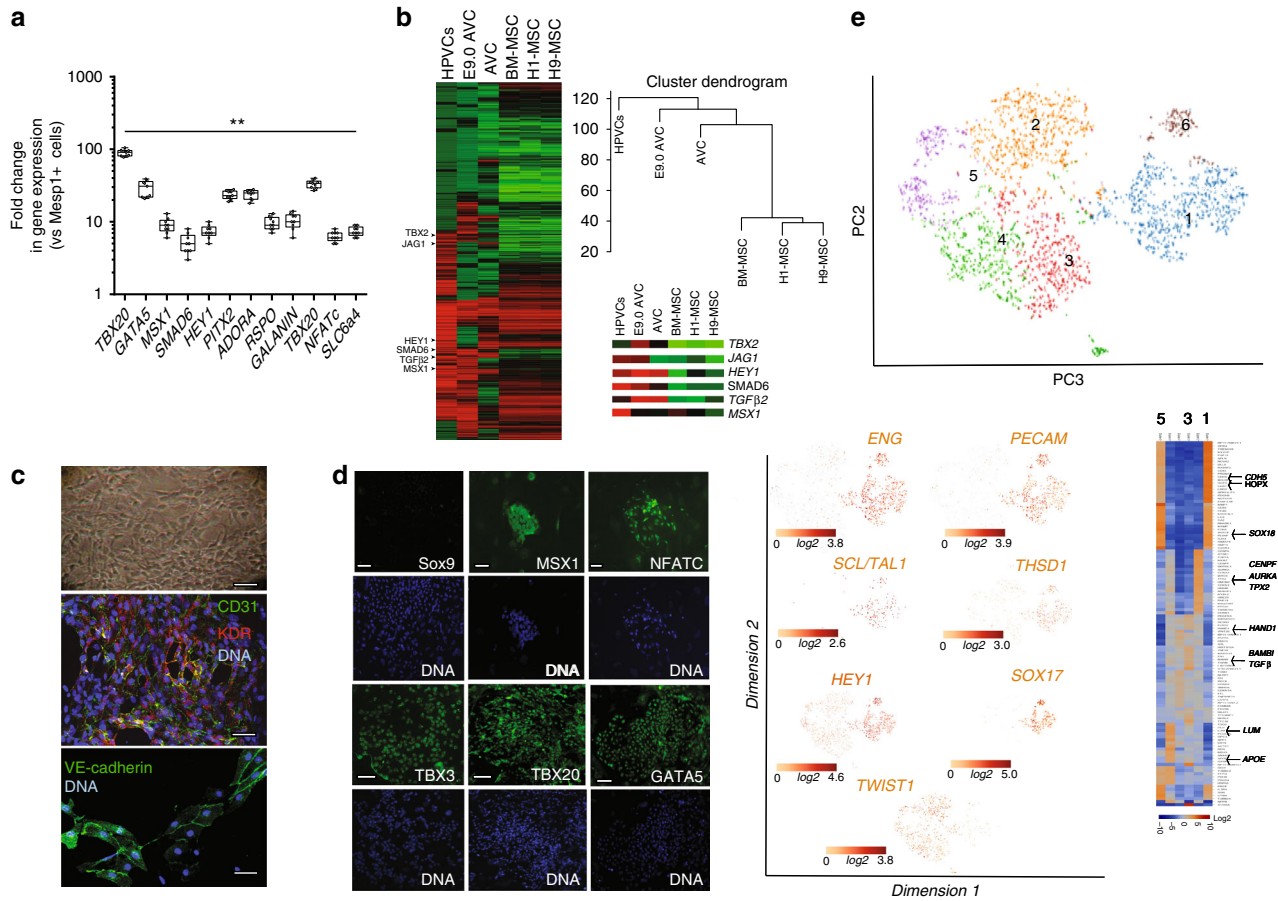

**Fig. 1** Gene and protein expression profiles of human valve progenitors. SSEA-1+-sorted human MESP1+ cells were treated with 10 ng/ml FGF8, 2 ng/ml FGF2 and 30 ng/ml VEGF for 6 days. **a** RNA was then collected and cDNAs of FGF8/VEGF-treated cells (HPVCs) were run in Real-Time PCR (mean ± SEM of 9 separate cell differentiation experiments). Data are normalized to 1 as the level of gene expression in SSEA1+MESP1+ cells. (**significantly different from 1; $p \leq 0.01$). Boxes and whiskers (min to max) show the values lower than the 2.5th percentile and greater than the 97.5th percentile as circles. **b** cRNAs ($n = 3$ separate cell differentiation experiments) were used for microarrays and normalized vs. MESP1+ cells from the same respective cell sorting. Heatmaps of transcriptomes of HPVCs, E9AVC (our data), AVC GDS3663 and MSCs (GDS1288). A few AVC-specific genes are highlighted in the inset. **c** Bright field image (top) and co-immunostaining of VEGF/FGF8/FGF2-treated SSEA-1+/MESP1+ derived colonies with anti-CD31 and anti-Flk1 (KDR) or anti-VE-cadherin antibodies. Data are representative of 5 separate cell differentiation experiments. **d** Immunostaining of VEGF/FGF8-treated SSEA-1+/MESP1+ derived colonies with anti-Sox9, -Msx1,-Nfatc1,-Tbx3, -Tbx20 and -GATA5 antibodies (green) and DAPI (blue). The data are representative of 5 experiments. The scale bar indicates 50 μm. **e** HPVCs were further sorted using anti-CD31 conjugated beads and used in single-cell RNA sequencing. t-distributed stochastic neighbor embedding (t-SNE) 2D cell map 10X genomics ($n = 2440$ cells) (upper panel). Highlight of cell populations expressing genes marking endothelial, hemogenic and early EMT cells (lower panel) and heatmap of graph-based Log2 fold changes in gene expression of cell cluster compared to all other cells (lower right panel). Source data are provided as a Source Data file

or bone marrow derived mesenchymal cells as well as with E9.0 AVC cells indicated HPVCs clustered with E9.0 AVCs and to a lesser extent to previously reported AVCs[28] and displayed little correlation with human ESC-derived or mesenchymal stem cells (Fig. 1b). The HPVC transcriptome further showed the presence of genes specific to AVC (*TBX2*) and endocardium (*MSX1*) as well as genes involved in Notch signaling (*JAG1, HEY1*) or in endocardial cushion mesenchymal transition (*TGFβ, SMAD6*) (Fig. 1b).

Immunofluorescence and high content imaging confirmed that HPVCs featured morphology of endothelial cells and showed expression of endothelial CD31 and KDR in 95% of cells (out of 850 scored cells in 2 separate experiments), VE-cadherin (Fig. 1c), as well as AVC and endocardial cells (GATA5, TBX3, TBX20, NFATc and MSX1) protein markers (Fig. 1d).

To estimate the cell heterogeneity of HPVCs, a single cell-sequencing approach was used. SSEA1+ MESP1+ sorted cells were plated on fibronectin-coated dishes and induced with VEGF/FGF8/FGF2 to an endocardial cell phenotype. Cells were then further sorted using an anti-CD31 antibody and used for single cell-sequencing. We performed a two-step data analysis to cluster cells by principal component analysis. Figure 1e revealed the endocardial phenotype of ~3000 HPVCs at a single cell level. Although the cells were CD31-sorted, the cell population was heterogeneous. Five main cell clusters were identified as expressing a specific gene pattern depending upon their stage in the early process of EMT (Fig. 1e).

One third of cells (1105) (cluster1) were *CDH5+* including endoglin (*ENG+*) cells (849), *PECAM1+* cells (729), *KLF2+* cells (161) as well as *NOTCH4+* (1005) cells, pointing to an endothelial and endocardial cell population. These cells also highly expressed *SOX17, SOX18, KDR* and *ETS* indicative of the endocardial phenotype (Supplementary Data 1). *TWIST1+* cells clustered as a mirror of *ENG+* cells although some cells were *ENG+TWIST1+*. 90% of *TWIST1+* cells were negative for *SNAI1* but still positive for *ETS1*, an endothelial gene confirming the early EMT stage of these cells. *THSD1* as well as *HEY1*, AVC endocardial genes were both enriched in the *CDH5+* cell

population (Fig. 1e) while *GATA5, TBX2, TBX3, MSX1* were found expressed in both endocardial and *TWIST1*+ cells (Supplementary Data 1).

Interestingly, a cluster of cells (180 cells within cluster1) expressed high level of *ETS1, EGFL7, HOPX* and *SCL/TAL1* suggesting the presence of a hemogenic endocardial cell population[29] (Fig. 1e). This cluster was dissociated from the small cell cluster 6 enriched in cells expressing *SOX18, LATEXIN, FICOLIN3, HOXP1, CARBONIC-ANYDRASE, CD40, CD55, CX37*,other markers of hematopoietic cells. Cluster 2 included highly proliferative cells expressing genes involved in cell mitosis such as *GTSE1, CENPF, AURKA, BIRC5* and *TPX2*. Cluster 3 included cells expressing genes of the TGFβ signaling pathway (*BAMBI, TGFβ1*) and *PDGFRα*. Cluster 4 was enriched in *WNT2B* cells but did not express any other genes not expressed in other clusters. Cluster 5 included cells more advanced in the EMT process expressing among others *APOE*, high level of *LUM, POSTN*, and *ACTA2*.

Thus, human ESC-derived HPVCs faithfully reflected the phenotype of endocardial cells before and at the onset of EMT within the AVC and showed robust correlation of gene expression with E9.0 AVC cells rather than with any other mesenchymal stem cell type.

**HPVCs undergo EMT and depend on Notch signaling in vitro.** To test the potential of hPSC-derived HPVCs to undergo EMT, a key process executed by endocardial cells to form cardiac cushions in vivo, HPVCs were further treated with BMP2 (200 ng/ml) for 48 h. Gene expression was monitored by real-time RT-PCR. When compared to non-stimulated cells, the level of expression of *MSX1, SMAD6, SOX9, SLUG, CADHERIN 11, N-CADHERIN AND PERIOSTIN* was significantly increased while *E-CADHERIN* was decreased (Fig. 2a).

Single cell-sequencing analysis was also performed after BMP2 treatment. Clustering within 5 tight groups revealed that most cells (2598) expressed TGLN, a myofibroblast marker enriched in the ventricularis layer of the valve[30]. Cells initiated specification into fibrosa, spongiosa or ventricularis layers of the valve. The *COL1A1* cell cluster (cluster 3) was enriched in most collagen genes (*COL1A1, COL1A2, COL4A1, COL6A2, COL3A1*) as well as in genes enriched in fibrosa such as *BGN*. Cluster 2 was enriched in genes *TDGF1, CD9, VCAN*, and *ALDH2* found in the spongiosa. Cluster 1 included endothelial cells still expressing *CDH5* and *PECAM1* (Fig. 2b; Supplementary Data 1).

Notch has a crucial function in the process of EMT in cardiac cushions[31,32]. We thus tested the role of the Notch pathway in BMP2-induced EMT of HPVCs. BMP2-induced expression of *SLUG* and *PERIOSTIN* was inhibited by 1 μM DAPT (N-[N-(3, 5-Difluorophenacetyl)-L-alanyl]-S-phenylglycine t-butyl ester) (Supplementary Fig. 4a), a γ-secretase inhibitor that blocks Notch pathway activation, indicating that expression of these two markers is Notch-dependent. Activation of the Notch pathway following transfection of Notch intracellular domain NICD strongly turned on the expression of *PERIOSTIN* as well as of *SLUG* and *MSX1* (Supplementary Fig. 4b), suggesting that as reported in vivo[31,32]. Notch regulates EMT via *SNAIL* and *SLUG* activation. Thus, HPVCs respond to similar cues and use equivalent signaling pathways to undergo EMT in vitro as mouse endocardial cells in vivo.

Valvular interstitial cells (VICs) give rise to tenocytes and osteo/chondrogenic cells[33,34]. We thus tested the tendinous/chondrogenic potential of HPVCs. We applied for 2 weeks a chondrogenic medium[33] to HPVCs aggregated in pellets, and found turned-on expression of *SCLERAXIS* and *COLLAGEN1* genes (Fig. 2c) as well as SOX9 and CALCITONIN proteins,

suggesting a broad valve differentiation repertoire of HPVCs (Fig. 2d).

**WNT stimulation of HPVCs upregulates KLF2 and EMT genes.** To test whether HPVCs could be at least in principle used in mechanostranduction experiments, we tested whether KLF2, a gene involved in the transcriptional response of hemodynamic forces[35] and expressed in a subset of endocardial HPVCs (Supplementary Fig. 5a), could be upregulated by Wnt stimulation. Freshly sorted CD31+ HPVCs were stimulated with 100 ng Wnt3a (expressed in HPVCs in the scRNAseq data) and 10 ng spondin3 for 24 h in ECGM medium. Supplementary Fig. 5b shows that cells turned on *AXIN2*, an index of Wnt response, and concomitantly strongly upregulated *KLF2*. Wnt also turned on *SLUG, MSX1*, and *SMAD6*, suggesting that cells underwent EMT.

**HPVCs undergo EMT in ex vivo models.** To further test the differentiation properties of HPVCs, we used collagen hydrogels to induce EMT. Cells were aggregated in hanging drops for 18 h generating spheroid bodies cultured on collagen gels. SSEA1-/MESP1- cells remained at the surface of the gel and proliferated. In contrast, HPVCs invaded the gel and acquired an elongated mesenchymal morphology (Fig. 3a). After 72–96 h, gels were fixed and cells immunostained to test expression of post-EMT markers. Figure 3b shows that human cells expressed FILAMIN A, PERIOSTIN, VERSICAN, AGGRECAN and smooth muscle actin (SMA).

Next, we utilized three-dimensional collagen substrate or heart explants[36] to test the behavior of HPVCs ex vivo. AVC regions of E9.5 mouse embryos were dissected and placed on a collagen hydrogel (Fig. 3c top inset). HPVCs were added to AVC explants for two days, fixed, and stained with anti-SOX9, -FILAMIN A or anti–PERIOSTIN antibody. Figure 3c shows that human cells identified by an anti-human LAMIN A/C antibody and in contact with endocardial cells of the mouse embryonic explant underwent EMT and expressed SOX9, FILAMIN A and PERIOSTIN. Thus, HPVCs are capable of recapitulating the EMT process in response to myocardial factors, likely BMP2 in an ex-vivo setting.

**HPVCs undergo EMT in vivo in mouse embryos.** To investigate the potential of differentiation of HPVCs in vivo, we tested whether these cells could further differentiate into valvular fibroblasts when placed in the embryonic environment at a stage of development prior to endoMT. To track HPVCs that have undergone Endo-MT in vivo in mouse embryos, a HUES cell line expressing GFP under the transcriptional control of the *Sox9* promoter was engineered. Sox9 was chosen as it is enriched in valves when the cushion endocardial cells were just at the beginning of the EMT process. HPVCs derived from the *Sox9*GFP HUES clone were injected into the AVC wall with injections mapped by fluorescent dye (Fig. 4a). Four to six hours after cell injection, embryonic hearts were dissected and cultured at the liquid-air interface in a matrigel coated insert for 48 h in DMEM plus 50% FCS. Hearts cultured under these conditions maintained their shape and beating activity for 72 h (Fig. 4b). Examination of sections of paraffin-embedded cell-injected hearts revealed activation of the Sox9 reporter in transplanted hESCs-derived HPVCs. Specificity of the enhancer/promoter was validated by a counterstain of GFP+ cells using an anti-SOX9 antibody. HPVCs stained by the anti-human LAMIN A/C antibody remained in the AVC region (Fig. 4b).

Alternatively, HPVCs derived from the *Sox9*GFP HUES clone were injected into the AVC of E9.5 embryos and embryos cultured in their yolk sac for 24–48 h. Hearts were allowed to

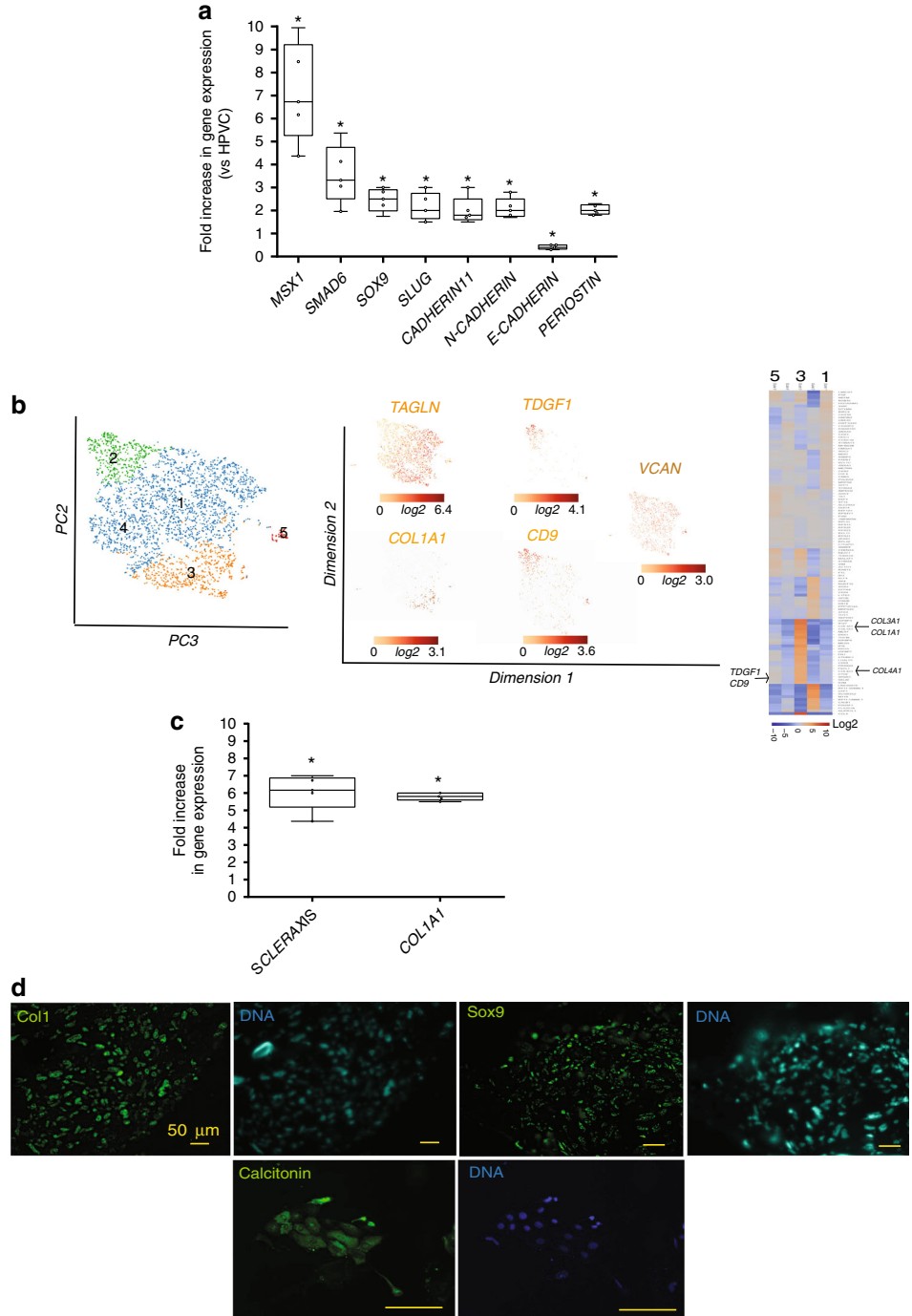

**Fig. 2** EMT of HPVC cells. **a** After 6 days of FGF8/FGF2/VEGF treatment on MEFs, valve progenitors (HPVCs) were recovered with trypsin, seeded on fibronectin-coated wells and treated with 100 ng/ml BMP2. After 2 days, RNA was recovered and cDNAs were run in real-time PCR for post EMT markers. BMP2 samples are normalized on control (before treatment) samples, showing an increase in the expression of post-EMT markers Data are representative of 5 cell differentiation and EMT- induction experiments.Boxes and whiskers (min to max) show the values lower than the 2.5th percentile and greater than the 97.5th percentile as circles. (*) significantly different $p \leq 0.01$. **b** t-distributed stochastic neighbor embedding (t-SNE) 2D cell map 10X genomics ($n = 3700$) cells (left panel). Highlight of cell populations expressing *TAGLN* and genes marking more specifically fibrosa (*COL1A1*) or spongiosa (*CD9, TDGF1*) (middle panel) and heatmap of graph-based Log2 fold changes in gene expression of cell cluster compared to all other cells (right panel). **c** HPVCs give rise to tendinous/chondrogenic cells. HPVCs were pelleted into a tube and treated with a chondrogenic medium as described in the methods. Real-Time PCR shows an upregulation of *Scleraxis* and *Collagen 1a* (*Col 1*) genes versus non-treated cells following three weeks of treatment (two experiments in duplicate; (*) significantly different $p \leq 0.01$). **d** The cell pellets were embedded in paraffin and 5 um sections stained with anti Sox9, anti-Collagen1a and anti-calcitonin antibodies. Data are representative of 3 experiments. The scale bar indicates 50 µm. Source data are provided as a Source Data file

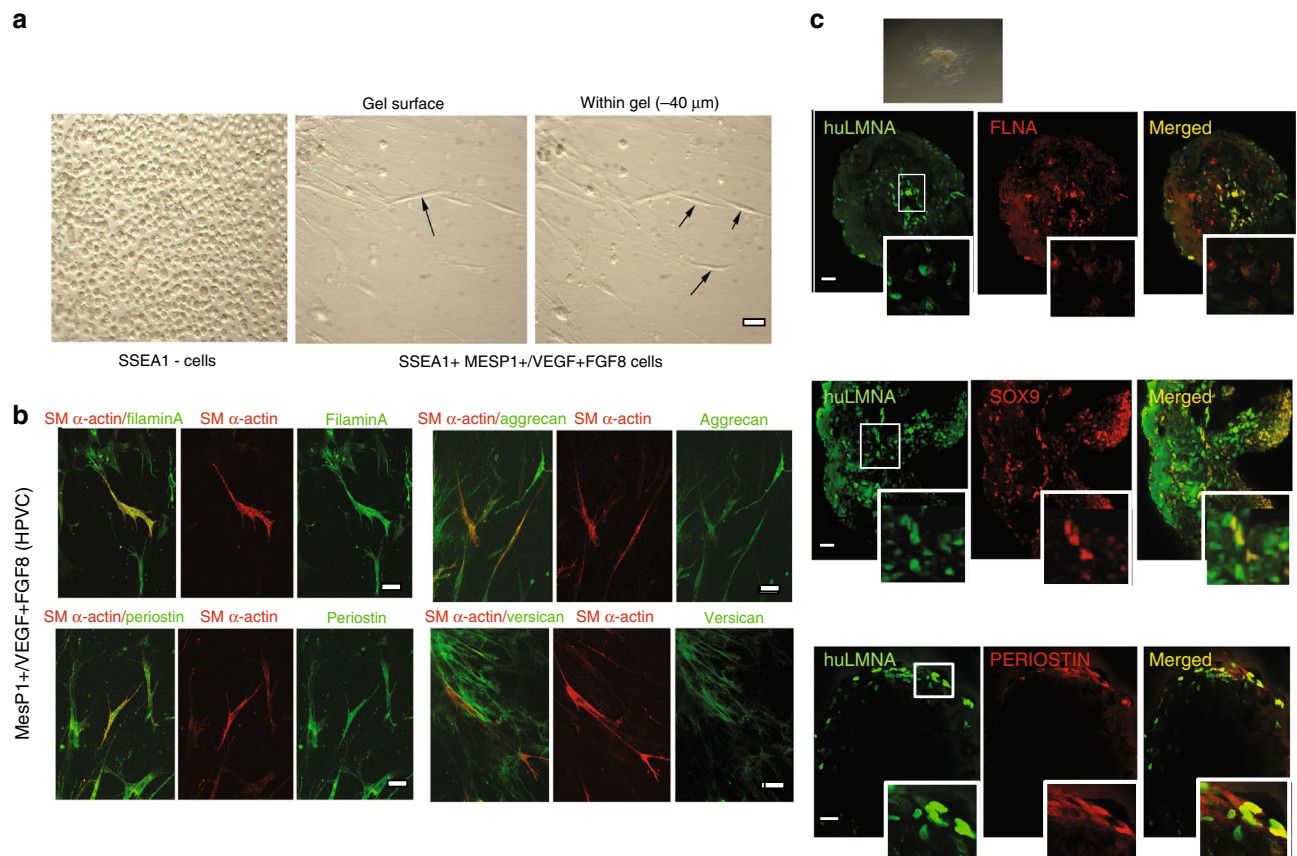

**Fig. 3** Ex vivo differentiation of HPVCs. **a** Left panel: After 4 days of BMP2 treatment MESP1-(SSEA1-) cells were collected and cultured in hanging drops. Cells were able to form aggregates, but when plated on collagen I hydrogels they did not adhere or invade the gel. Middle and right panel After SSEA1+ sorting and treatment for 6 days on MEFs with FGF8, FGF2 and VEGF, 40,000 valve progenitors were cultured as hanging drops for 18 h and plated as aggregates on collagen I hydrogels. In 48 h, invasive mesenchymal cells were seen in the gel (−40 μm from the surface), **b** A total of 20,000 MESP1+, or MESP1- progenitors or FGF8/FGF2/VEGF-treated MESP1+ progenitor cells were mixed with 20,000 freshly isolated chick H-H stage-24 AV cushion mesenchymal cells to aggregate. Cells were cultured with or without BMP2 for 48 h and stained for post-EMT markers SM alpha actin together with filamin A, periostin, versican, and aggrecan. Very few SSEA1+ and SSEA1+− cells adhered and survived, whereas FGF8/VEGF-treated cells compacted themselves at the middle of the chicken-human cell aggregate and, when triggered with BMP2, some cells started to spread out from the aggregate. **c** FGF8/VEGF-treated MESP1+ progenitors were placed on the top of E9.5 mouse AVC explants on collagen gels (inset), cultured for 2 days and co-immunostained with anti-human LaminA/C and anti-Sox9, -filamin A or -periostin antibodies. Bar scale indicates 50 μm. Data are representative of at least 6 separate experiments

develop in culture (Fig. 4c). The HPVCs injected in the chamber or ventricular wall or in an extra-cardiac environment (brain) survived poorly and did not express GFP, SOX9, PERIOSTIN, FILAMIN or COLLAGEN (Fig. 4c). Immunostaining of Sox9GFP cells revealed that cells implanted in the AVC expressed PERIOSTIN, FILAMIN A AND COLLAGEN I (Fig. 4d). At high magnification, Sox9GFP cells were surrounded by extra-cellular matrix proteins PERIOSTIN and COLLAGEN. FILAMIN A (FLNA), which functions as a scaffolding protein and couples cell cytoskeleton to extracellular matrix, was also located at the periphery of the cells (Fig. 4d, right panel). *Sox9*GFP HUES cell-derived HPVCs injected in the E9.5 AVC were also found in the AVC both in the superior and inferior atrio-ventricular cushions and expressed PERIOSTIN following 36 h culture of whole embryos (Fig. 4e).

**Patient-specific iPS cells recapitulate MVP**. To test the patho-logical relevance of HPVCs and derived valvular cells, we used a patient-specific iPS cell model. iPS cells were derived from valv-ular interstitial cells (VICs) isolated from the explanted mitral valve of a patient harboring a mutation in Dachsous gene (*DCHS1*) a likely cause for mitral valve prolapse[9]. In contrast to

primary VIC culture, which is limited by the number of passages, iPS cells provide a means to recapitulate the stepwise determi-nation of distinct endocardial, valve endothelial, and valve interstitial cell phenotypes. HiPS cells allow replication of sepa-rate differentiation experiments both with HPVCs/VECs and VICs. Patient-derived hiPs cells expressed OCT4, SOX2 and NANOG (Fig. 5a). DNA sequencing confirmed that the patient-specific *DCHS1* mutation *c-6988C>T* was conserved (Fig. 5b) and digital PCR of copy number variants of main recurrent abnormalities loci found in pluripotent stem cells (http://www.stemgenomics.com/) revealed that genomic integrity was con-served. Using directed differentiation protocols, cells were capable of differentiation into the three germ layers following treatment with specific morphogens as shown by expression of *SOX17*, and *FOXA2* (endoderm), *BRACHYURY (T)* and *MIXL1*, (mesoderm), and *NESTIN* and *PAX6* (ectoderm) (Fig. 5c).

Dachsous is involved in planar cell polarity signaling and in turn cilia formation[37] and is required for lymphatic valve formation[38] Valve endothelial cells and their human counterpart (i.e. hiPS cell-derived HPVCs) expressed endocardial genes TBX2, TBX20, PITX2, GATA5, SMAD6 and MSX1 (Supple-mentary Fig. 6), but expressed few cilia as recently reported in

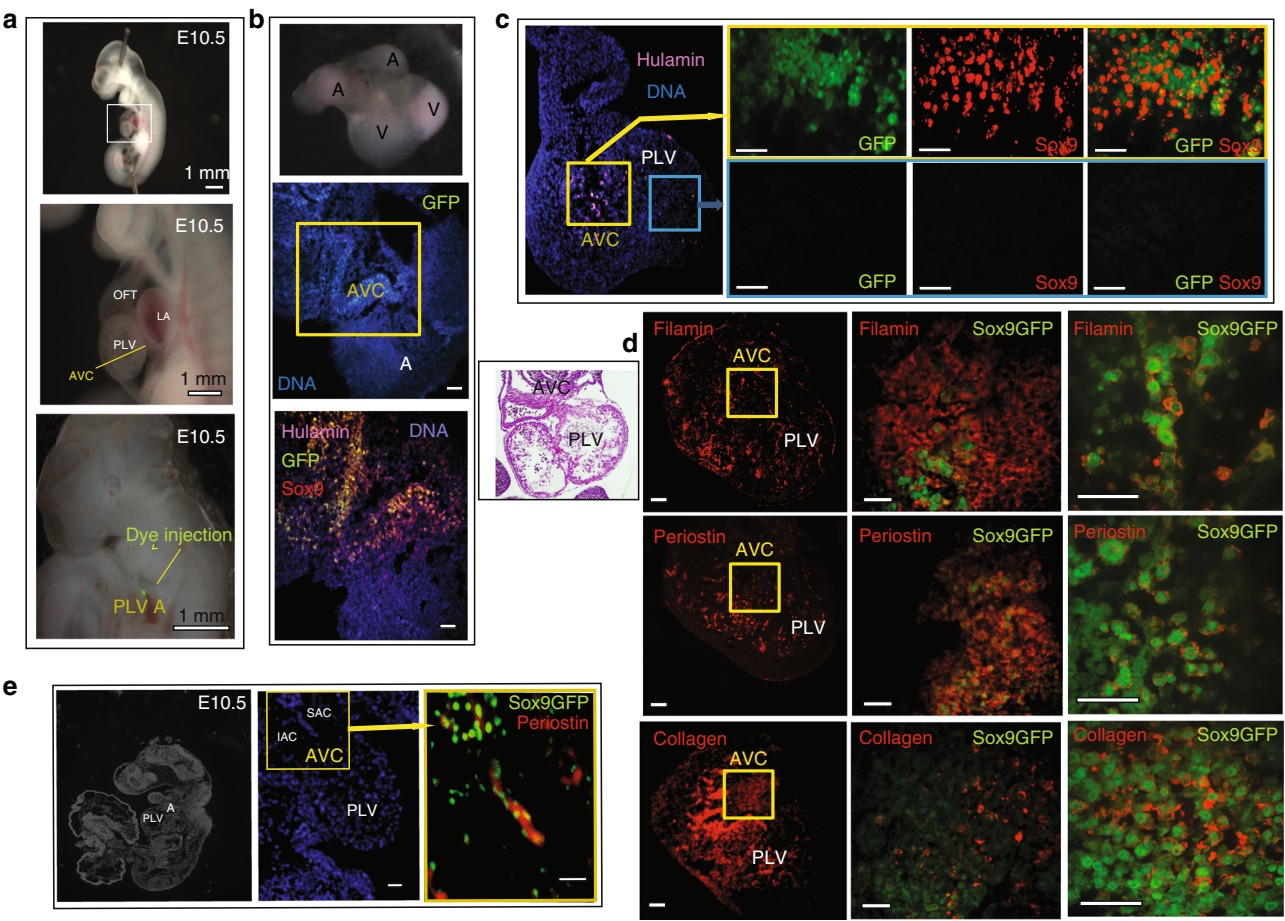

**Fig. 4** In vivo differentiation of HPVCs in mouse embryonic heart. SSEA1+ MESP1+ Sox9GFP cells were sorted and treated with VEGF/FGF8 for 7 days before injection into E10.5 mouse AVC. **a** An E10.5 embryo is stretched to expose the AVC. Dye injection shows the target of the injection. Scale bars are 1 mm. **b** Two hours after injection the hearts were dissected and placed in culture for two days on Matrigel coated inserts (top inset). The heart sections were immunostained with anti-human LmnA/C, –GFP, and –Sox9 antibodies. **c** Two sites of injection and resulting cell fate. Cells were injected into the AVC or the chamber. Only the cells injected into the AVC (visualized on the eosin/hematoxylin stained heart section in the inset) underwent EMT and expressed Sox9-GFP. **d**, **e** Sox9GFP VEGF/FGF8-treated cells were injected into the AVC and the hearts were cultured for two days (**d**) or the whole embryos were cultured in their yolk sac for 36 h (**e**).The heart sections were stained with anti-filaminA, -periostin or –collagen antibodies. The middle panels in (**d**) show the AVC region and the right panels a high magnification of a subset of cells within the AVC (**e**). A section of the whole embryo is shown in the left panel; the middle and right panels show the heart and the cells injected in the AVC. The heart sections were stained with anti-periostin antibody. Results are representative of at least 9 experiments. SAC superior AV cushions, IAC inferior AV cushions, PLV primary left ventricle, A atrium. The scale bars in b-e indicate 50 μm

mouse[39] potentially due to the high shear stress in the region of cardiac cushions as proposed in zebrafish[40]. EMT was thus induced by BMP2 treatment of HPVCs and cilia were scored in VIMENTIN+ VICs (Fig. 5d). 450 HPVCs derived VICs stained with anti-acetylated -α-tubulin revealed that 56% of wild type (wt) cells while only 12% of DACHS1 c.5988 C>T cells featured cilia. The identity of cilia was confirmed by counterstaining with an anti−γ-tubulin antibody (insets Fig. 5d). Furthermore, cilia of mutated cells were twice as short as the ones of wild type cells (Fig. 5d).

Mitral valve prolapse as a consequence of *DCHS1 c.6988 C>T* mutation leads to a myxomatous degeneration and an increase in proteoglycan and collagen I expression compared to wild type healthy valve[6]. Wild-type and *DCHS1 c.6988 C>T* HPVCs derived interstitial cells were thus stained with the HYALURONAN-binding protein to visualize HYALURONANand an anti-COLLAGEN I antibody. Figure 5f revealed that mutated cells secrete three times as much COLLAGEN I (22.7 ± 3 % of cell field, $n = 5$ experiments, 540 scored cells) than wild type cells (7.4 ± 1.2 %) and six times as much HYALURONAN (36 ± 6%, of

cell field $n = 5$ experiments, 645 scored cells) than wild-type cells (5.7 ± 1.5 %). They also featured 7 times as much PERIOSTIN (28 ± 4%, $n = 5$ experiments, 530 scored cells) than wild type cells (4 ± 0.8%).

**SHH signaling is altered in *DCHS1 c.6988 C>T* HPVC-derived VICs.** Sonic hedgehog (SHH) is a signaling pathway of primary cilia. SHH regulates expression of HYALURONAN[41], and play key roles in ECM remodeling in development. SHH receptor Patched1 is located on or at the basal bodies of cilia and relocates to the entire cell membrane when bound by its ligand where it gets activated even without ligand[42]. We reasoned that *DCHS1 c.6988 C>T* valvular cells with short or no cilia may feature a constitutive activation of SHH pathway, following a lack of Patched1 on cilia. We first looked at the location of PATCHED1 in wt and *DCHS1 c.6988 C>T* cells. Figure 6a shows that Patched1 was clustered and aggregated at the cilia basal body in wt cells while it was spread as clusters all over the cell membrane in non-ciliated *DCHS1 c.6988 C>T* cells.

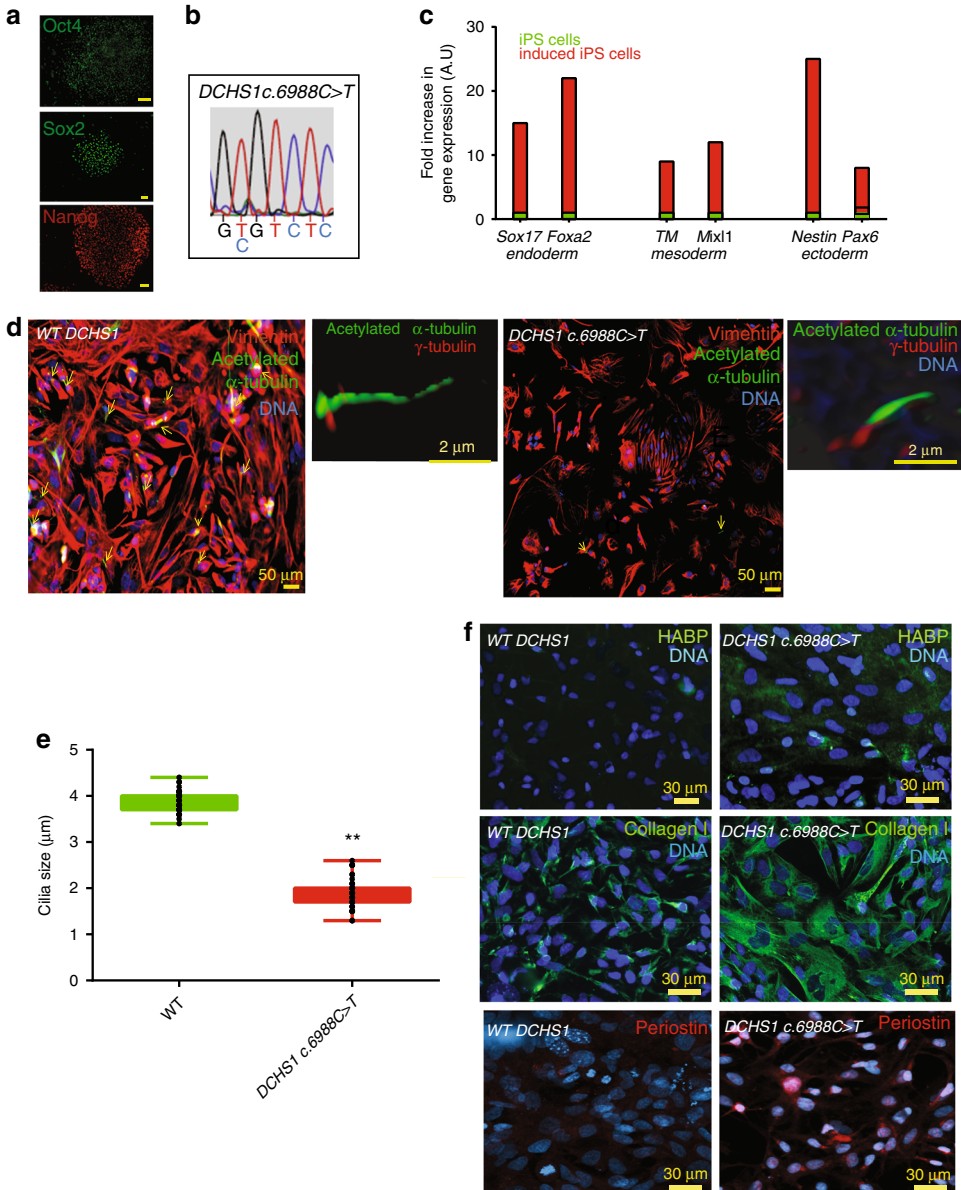

**Fig. 5** DCHS1 c.6988 C>T iPs cells recapitulate the patient valvular phenotype. **a** iPS cells derived from mitral valvular interstitial cells express OCT4, SOX2 and NANOG. **b** The patient-specific *DCHS1 c.6988 C>T* mutation was conserved following iPS cell derivation. **c** *DCHS1 c.6988 C>T* carrying iPS cells were differentiated toward the endoderm, mesoderm and ectoderm. Real-time PCR of Sox17, Foxa2 (endoderm), Brachyury (T) and Mixl1 (mesoderm) and Pax6 and Nestin (ectoderm) confirmed multilineage propensity. **d** Wild type (WT DCHS1) or mutated (DCHS1 c.6988 C>T) HPVCs underwent EMT under BMP2 stimulation and were immunostained with anti-vimentin and α-acetylated tubulin to visualize cilia. Insets show 3D-image of cilia co-immunostained with both anti-α-acetylated tubulin and anti-γ-tubulin antibody. **e** Quantification of cilia size: Cilia were stained by anti-α-acetylated tubulin and anti-γ-tubulin. Length of the cilia was measured using Image J in 47 cells randomly chosen from 3 separate experiments. Boxes and whiskers (min to max) show the values lower than the 2.5th percentile and greater than the 97.5th percentile as circles. (**significantly different, $p \leq 0.01$). **f** Hyaluronan, collagen I and periostin were visualized by anti-HABP antibody, anti-collagen I and anti-periostin antibodies, respectively in cultures of VIC derived from both wild type (WT *DCHS1*) or mutated (*DCHS1 c.6988 C>T*) HPVCs. To quantify HYALURONAN and collagenI, 3 fields in 3 separate experiments were scored using the image thresholding mode of Image J. Source data are provided as a Source Data file

We thus treated *DCHS1 c.6988 C>T* cells with cyclopamine, a hedgehog pathway inhibitor, together with BMP2 at the onset of EMT. The drug prevented in a significant manner ($p \leq 0.01$) secretion of excess collagen I (from $25 \pm 3$ down to $2 \pm 0.3\%$ cell field, $n = 3$, 520 scored cells) HYALURONAN (from $30 \pm 2.1$ down to $1.65 \pm 0.5\%$ cell field, $n = 3$, 480 scored cells), or PERIOSTIN (from $20 \pm 1.7$ down to $1.5 \pm 0.2\%$ cell field, $n = 3$, 515 scored cells) in *DCHS1 c.6988 C>T* cells (Fig. 6b). We next tested whether SHH added to wt cells could mimic DCHS1 c.6988 C>T phenotype. Addition of 100 ng/ml SHH for 48 h on

wt cells treated with BMP2 increased in a significant manner ($p \leq 0.01$) the expression of COLLAGEN I (from $6.8 \pm 1$ up to $16.7 \pm 2.1\%$ of cell field, $n = 3$, 340 scored cells), HYALURONAN (from $10 \pm 1.2$ up to $17.1 \pm 1.9\%$ of cell field, $n = 3$, 380 scored cells) and PERIOSTIN (from $4 \pm 0.7$ up to $31.2 \pm 2.7\%$ of cell field, $n = 3$, 490 scored cells) (Fig. 6c). In contrast to wild type cells (Fig. 7), single cell sequencing of *DCHS1 c.6988 C>T* post-EMT cells revealed that cells were dramatically heterogeneous as illustrated by the t-SNE graph (Fig. 7a) and the heatmap of cell cluster (Fig. 7c). A principal component analysis of WT vs *DCHS1*

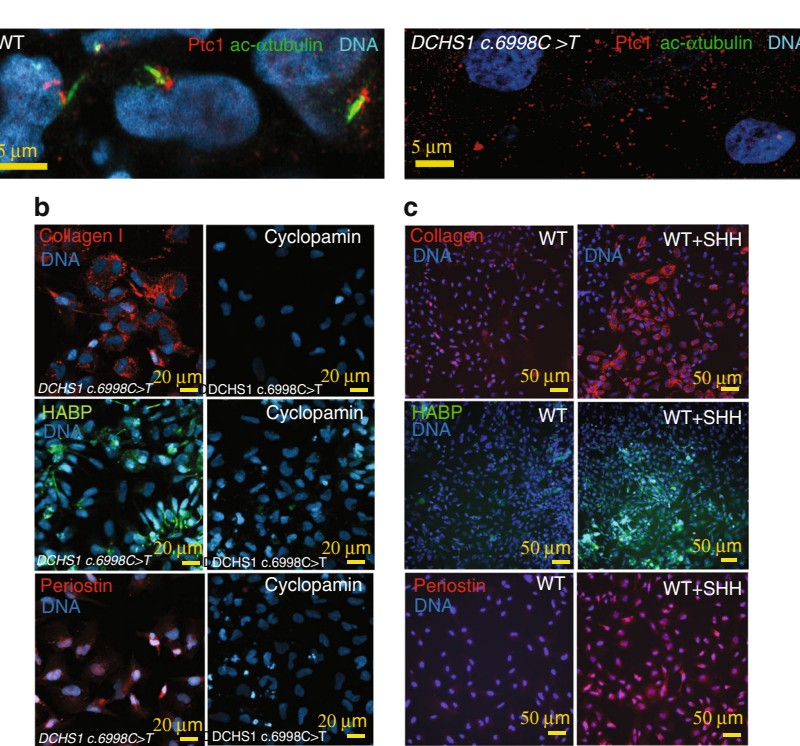

**Fig. 6** SHH over-activation in mutated (*DCHS1 c.6988 C>T*) cells. **a** SHH receptor patched I (Ptc1) and the cilia (anti-acetylated α−tubulin) were co-stained on both ciliated wild type (WT) or unciliated *DCHS1 c.6988 C>T* cells. The confocal images are representative of 3 separate experiments and 180 scored cells in each group. Hyaluronan, collagen I and periostin were visualized by anti-HABP antibody, anti-collagen I and anti-periostin antibodies, respectively in cultures of VIC derived from both mutated (*DCHS1 c.6988 C>T*) or wild-type (WT *DCHS1*) HPVCs treated with cyclopamine (**b**) or SHH (**c**), respectively. To quantify collagen I, hyaluronan and periostin, 3 fields in 5 separate experiments were scored using the image thresholding mode of Image J

*c.6988 C>T* single VIC transcriptome revealed a distinct pattern of gene expression (Fig. 7b). Differences between WT and *DCHS1 c.6988 C>T* cells could not be attributed to a differential cell cycle as most cells of the two populations were in the G1 stage (inset Fig. 7b).

Patient-derived VICs could not acquire a typical VIC identity. All expressed high levels of *TAGLN, COLLAGEN I (COL1A1), VERSICAN (VCAN)* and *PERIOSTIN* (Fig. 7). Some genes enriched in a specific valve layer such as *TDGF1* (fibrosa) were not expressed at all in *DCHS1* mutated cells or like *CD9, APOE, BGN* (fibrosa) or *ACADM* (spongiosa) spread all over cell clusters. A small cluster (cluster 6) of cells was made up by endothelial cells highly expressing *PECAM1, CDH5, ENG, SOX17, SOX18, DLL4, CD34* and *KDR* (Fig. 7c) (Supplementary Data 1).

A comparison between WT and *DCHS1* mutated cells confirmed that the latter overexpressed ECM proteins of the fibrosa or spongiosa (*VCAN, COL1A1, PSTN, APOE, TAGLN, BGN, ACADM*) as shown by the heatmap and violin plots (Supplementary Fig. 7).

Interestingly GLIs 2 and 3 were also overexpressed (9.3 ± 1.1 fold in 3 real-time RT_ PCR experiments and in single cell data sets of DCHS1 c.6988 C>T vs wt cells (see Supplementary Fig. 8 and Supplementary Data 1). Furthermore, BRD2 and PRTM1, two positive regulators of SHH[43] were highly upregulated and more broadly expressed in DCHS1 c.6988 C>T vs wt valvular cells (Supplementary Fig. 8).

Interestingly GLIs 2 and 3 were also overexpressed (9.3 ± 1.1 fold in 3 real-time RT_ PCR experiments) and in single cell data sets of *DCHS1 c.6988 C>T* vs. wt cells (see Supplementary Data 1 and Supplementary Fig. 8).

Furthermore, *BRD2* and *PRTM1*, two positive regulators of SHH[43] were highly upregulated and more broadly expressed in DCHS1 c.6988 C>T vs wt valvular cells (Supplementary Fig. 8).

## Discussion

The present study documents the propensity of pluripotent stem cell-derived MESP1+ cardiovascular progenitors to give rise to bona fide HPVCs, which in turn provide a reliable source for functional signal-responsive valvular cytotypes, namely valvular interstitial cells and tendinous/chondrogenic cells. Establishment of pluripotent stem cell-derived HPVCs offers a powerful prototype platform of human early valvulogenesis addressing a major gap in this field. The ensuing application of patient-specific hiPS cell-derived valvular cells to model valvulopathy exemplifies a genuine model of nonsyndromic mitral valve prolapse, underscoring the biological validity and clinical utility of this proof-of-concept study.

To map the gene expression signature for the HUES-cell-derived valvular progenitor cell population (HPVC), we performed gene expression profiling on one of the regions of the heart where the valves form. We used E9.0 embryos to determine the transcriptomic identity of AVC prior to EMT, which occurs between E9.5 and E11. We show here that HPVCs derived from pluripotent stem cells accurately reflect the in vivo E9.0 AVC profile. The gene profile of AVC was compared to the adjacent primary chamber myocardium and endocardium, thus restricting the cell gene signature to the endocardium of the AVC. While the primary ventricle may feature to some extent a different gene profile than the AVC myocardium, our primary ventricle gene array did not show expression of CKM, a gene not expressed in

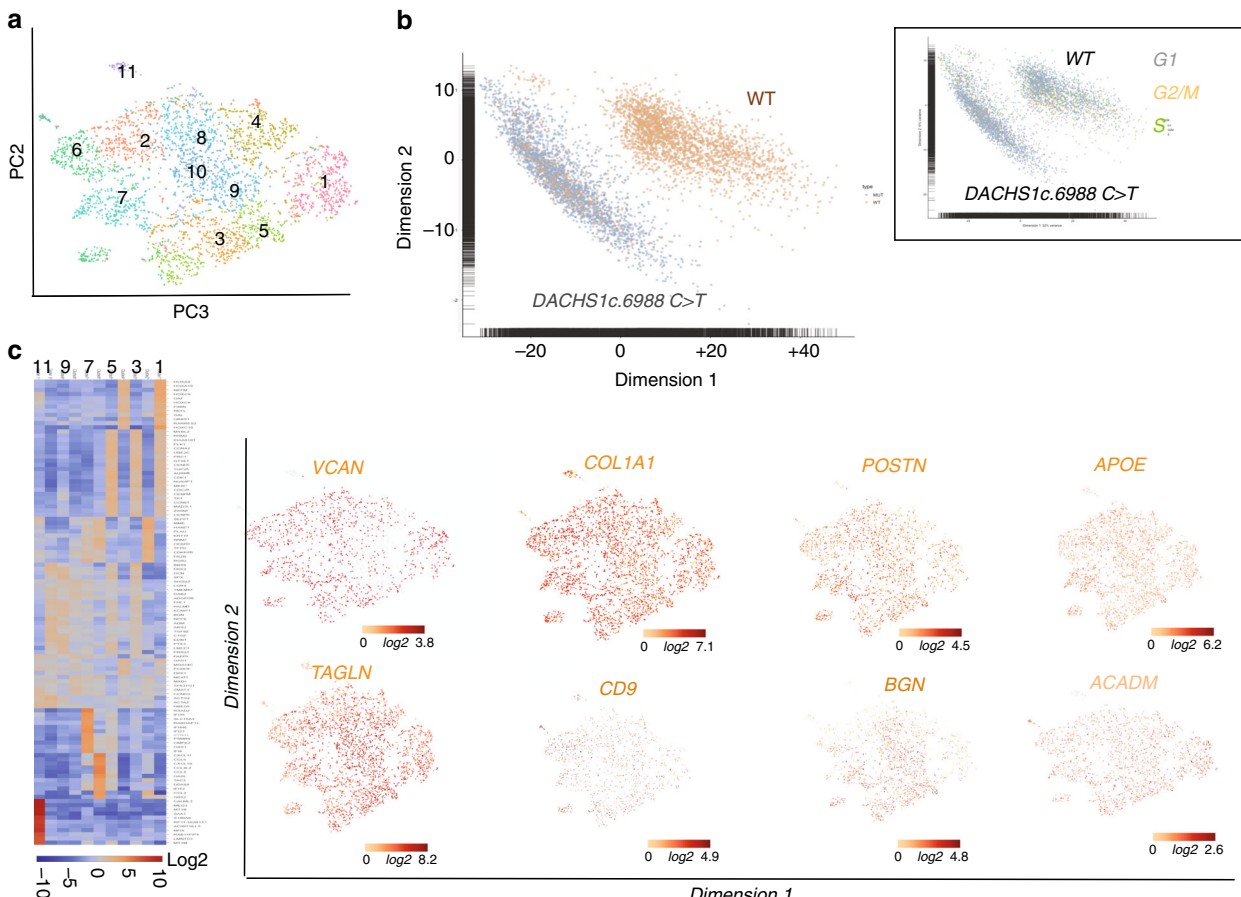

**Fig. 7** Single cell-sequencing of *DCHS1 c.6988 C>T* post-EMT valvular interstitial cells. **a** t-distributed stochastic neighbor embedding (t-SNE) 2D cell map 10X genomics (n = 4316 cells). **b** PCA of wt versus *DCHS1 c.6988 C>T* post-EMT (BMP2 treated) HPVC cells. Inset: PCA of cell cycle genes expressed in wt versus *DCHS1 c.6988 C>T*. **c** Graph-based Log2 fold changes in gene expression of cell clusters compared to all other cells (left panel) and highlight of cell populations expressing genes marking fibrosa or spongiosa valve layers

AVC myocardium in contrast to mature chamber myocardium[44] pointing to a still immature myocardium. Gene expression arrays of this cardiac-restricted region (i.e., using Tbx3 GFP sorted cells from E10.5 mouse embryos) so far available in the literature were done at a later stage of development, thus during EMT, and included both endocardium and myocardium[45]. An attempt to carry out a transcriptome of E9.5 AVC was reported previously[28]. However, a careful analysis of these data revealed that the AVC transcriptome was contaminated by high expression of chamber specific genes such as *TnT*, or *Gja1* (see GEO data set GDS3663) and key AVC specific genes such as *Tbx2* were missing (see GEO dataset GDS3663 and Fig. 1d). This could explain why this previously reported gene signature did not cluster tightly with those of our dissected AVC and HPVCs (Fig. 1e). One study using a SAGE protocol reported a gene expression profile of mouse cardiac AVC at a slightly later stage (E9.5) of development[45]. The comparison of our microarray data with that of the Tbx3-GFP-sorted cells[44] and the SAGE data[45] further revealed the expression of common and major genes in AVC vs. cardiac chambers. These include *Tbx2, Tbx3, Tbx20, Msx*, and *Id2*. Interestingly, four of these genes (*TBX3, TBX2, TBX20*, and *MSX*) were highly expressed in the HPVCs (Fig. 2). However, our array identified additional endocardial-specific genes, like *GATA5*, or *NFATC*[46] as well as genes restricted to AVC such as *TGFβ2*, and *HEY1*[32], which were also expressed in the HPVCs, including at a single cell level (Fig. 1). Genes suggestive of the cardiac cushions such as *Smad6, Twist*, endocardial *Msx*, a serotonin receptor variant

(*Slc6a4*)[47] were also found enriched in AVC endocardium vs chambers. Furthermore, we uncovered genes highly enriched in the AVC that were not previously reported to play a role in AVC identity or more specifically in cushion formation were. These include *caveolin* (*Cav1*), specifically expressed in the endocardium (Fig. 1) *thrombospondin* (*Thsd2*), the Wnt modulator *R-Spondin* (*Rspo3*), the adenosine receptor (*Adora*) and the neuropeptide galanin (*Gal*)[48]. Expression of most of these genes was also found in the HPVC population (Fig. 2). None of the genes found enriched in expression in the chambers (Nppa, Irx2) were present in AVC, nor in the HPVCs. Single cell RNA-sequencing further confirmed previous data. At a single cell level, endoglin (*ENG*) an endocardial-specific gene, VE-Cadherin (*CDH5*), KDR and *PECAM1, THSD1* were found highly enriched at different levels either in TWIST1-negative or -positive cells. *GAL, ADORA1*, and *CAV1* were all expressed at a single cell level.

The gene profiles of HPVCs as well as AVC were quite different from the ones of either the inferior or superior outflow tract. None of the mainly enriched genes in both AVC and HPVCs (*Cav1, Thsd1, Spo3, Vsnl1, Gal, Igfbp7, Shisa2, Tbx2, Hey1, Wnt2, Pitx2*) were expressed in OFT regions[49] thus pointing to a specific AVC signature of HPVCs.

These findings collectively point to the pre-valvular endocardial identity of the human cell-derived HPVC population and validate the protocol used to derive such a cell population. Overall, the transcriptome of HPVCs clustered with the one of

E9.0 mouse AVC and not with mesenchymal cells (Fig. 1), further emphasizes the identity of the HPVC. Together, these data reveal the unique potential of our HPVC model to discover and further investigate regulation of human valvulogenesis.

Our rationale to direct the fate of MESP1+ cardiovascular progenitor towards the myocardium and endocardium, and further to the endocardial cushions[50] and a valvular fibroblastic as well as tendinous/chondrogenic phenotype, was that these progenitors should acquire an endothelial identity and an endocardial cell fate within a cell population at the expense of myocardial outcomes, in line with in vivo valvular prioritization. We have found that FGF8, known to favor endocardium vs myocardium[24] as well as to play a role in formation and EMT of cardiac cushion[51,52], was potent in recapitulating the same cell determination process from pluripotent stem cell-derived MESP1 + cells[23]. We thus combined the action of VEGF, an endothelial inducer, with FGF8 driving mesodermal cell fate toward endocardial cells at the expense of myocardial lineage[24]. Stimulation by VEGF and FGF8 were combined with the influence of an extracellular fibronectin matrix and MEFs, which secrete FGF2 and TGFβ2[53] known to synergize with VEGF to determine the endocardium[54]. Together, this inductive signaling induced the HPVCs to express endothelial (i.e CD31, VE-CADHERIN, ENDOGLIN) and endocardial lineage genes as found in the AVC (i.e. GATA5, TBX2, GALANIN, RSPO, SMAD6, MSX1). This suggests that HUES cells can not only recapitulate the mesodermal cardiogenic pathway[23] but can also be directed toward the fate of endocardial pre-valvular AVC cushion cells (Fig. 6). These findings correlate with embryonic development as MESP1+ cells are known to give rise to endocardial cells and cardiac cushions in the mouse embryo[19,55].

We further show that human valve progenitors are capable of EMT under different experimental conditions. First, when cultured on collagen gels, HPVCs acquire a mesenchymal phenotype and express valvular fibroblast markers such as FILAMIN, PERIOSTIN, VERSICAN and AGGRECAN. Second, when cultured ex vivo or grafted in vivo with or close to cardiac cushions of the AVC in mouse embryos, respectively, only HPVCs, but not SSEA1- or SSEA1+ MESP1+ cells, become valvular interstitial cells expressing FILAMIN A, PERIOSTIN, SOX9 and COLLAGEN I, indicating that the HPVCs are unique and selective in their response to cues derived from the surrounding mouse AVC tissue.

The ex and in vivo EMT process is likely triggered through the secretion of BMPs by the myocardium, as indicated in in vitro mouse AVC explant cultures[56] and in genetically altered in vivo mouse systems[19,57]. Such a phenomenon could be recapitulated in vitro using BMP2, WNT and NOTCH signaling pathway activation with HUES cells-derived HPVCs. Indeed, BMP2 also induces expression of post EMT genes such as SLUG, SMAD6, PERIOSTIN, or SOX9. WNT also turned on expression of EMT genes including SLUG, SMAD6, and MSX1. Expression of SLUG, a direct notch target in vitro and in vivo[32,56], and PERIOSTIN were abrogated by the Notch inhibitor, DAPT. PERIOSTIN, SLUG and MSX1 expression was further induced by ectopic expression of the intracellular Notch domain (Supplementary Fig. 2). Interestingly, KLF2 was expressed in a subset of PECAM+, ENG+, CDH5+ HPVCs and was significantly upregulated by Wnt signaling (Supplementary Fig. 5), a response that may play a role in mechanotransduction by securing KLF2 expression via a positive feedback of Wnt signaling[35].

These findings thus confirm that maturation of stem cell-derived HPVCs recapitulates in vitro, at least partially, the in vivo embryonic signaling mechanisms mediating EMT and cell invasion, including the interplay between the Bmp2/Notch signaling pathways[32].

Primary ciliary dyskinesia (PCD) without situs inversus[58] as well as other ciliopathies have been clinically associated with myxomatous mitral valve and other valve diseases[59]. Interestingly, patients with autosomal dominant polycystic kidney disease (ADPKD), a disease in which cells feature an over-activation of the SHH pathway[60], have an increased occurrence of mitral valve prolapse[59]. Dachsous is required for lymphatic valve development[37]. Dachsous mutated HPVCs recapitulated several features of mitral valve prolapse (Fig. 5). This included cilia defects responsible for disorganization of valve interstitial cells within the leaflets and excess secretion of hyaluronan, collagen I and periostin, a hallmark of mitral valve prolapse.

HPVC single cell-sequencing data uncovered that DCHS1 c.6988 C>T valve interstitial cells could not acquire a normal VIC identity or express high levels of tissue-specific COL1A1 or PSTN. This lack of cell identity could explain why the cells cannot properly distribute within valve layers. This also suggests that ECM may regulate cell fate of valve interstitial cells in the course of leaflet patterning. Interestingly, cilia signal through sonic hedgehog, which targets the hyaluronic acid synthetase (Has) gene[40] and other components of the ECM[42] via the transcription factor Gli3. In the absence of sonic hedgehog (SHH), the GLI proteins GLI2 and GLI3 are phosphorylated by PKA, CKI and GSK3β. This leads to their proteolytic cleavage to generate repressor forms (GLI2R and GLI3R, respectively)[61]. A deregulation of this signaling pathway would relieve inhibition on Has and upregulate hyaluronic acid. Besides the role of cilia as a sensor of the extracellular environment (i.e. EC Matrix), the direct regulation of ECM organization may explain how mutations in Dachsous and the absence of cilia deregulate synthesis of ECM proteins through SHH pathway. Indeed, the SHH receptor patched1 was found clustered at the basal body of cilia in wild type cells while it was spread across the cell membrane in DCHS1 c.6988 C>T cells lacking cilia (Fig. 6). Furthermore GLIs 2 and 3 as well BRD2 and PRTM1, positive regulators of SHH targeted genes[43] were upregulated in DCHS1 c.6988 C>T cells valvular cells vs wild type cells, pointing to an activation of SHH pathway. Interestingly, a lack of cilia favors EMT in mouse Tg737/ift88 mutants[62], which is in agreement with our observation that DCHS1 c.6988 C>T undergo EMT like wildtype cells. However, DCHS1 c.6988 C>T mutant cells featured an over-activation of the SHH pathway in contrast to Tg737/ift88 mutant hearts[62], which lacked SHH signaling at least in second heart field progenitors.

Our data using a SHH inhibitor (Fig. 6) confirmed that SHH signaling is constitutively activated in DCHS1 c.6988 C>T cells. Regulation of the pathway by a pharmacological SHH pathway inhibitor rescued the ECM cell phenotype. Thus, our data bring a potential pharmacological target to potentially prevent MVP.

In summary, we report that (patient) hiPS cell-derived HPVCs can be used as faithful models for human valve disease.

A limitation of our study was the use of SSEA1+ MESP1 +sorted cells. Such sorting may have excluded the endocardial cells that are already determined in the primitive streak[13,15,63] and segregated from the myocardial lineages in the pre-cardiac mesoderm prior to the segregation of the first and second myocardial lineages[14,54], as reported in chicken, quail, and mouse embryos. Whether at least some endocardial cells are already determined in the pre-cardiac human mesoderm, remains to be further investigated.

Our in vitro, ex vivo and in vivo assays facilitate further studies examining the events that give rise to the human endocardium, valvular interstitial cells, and tendinous/chondrogenic cells, as well as studies to delineate the pathways triggering EMT and potentially underlying mechanotransduction. For such an aim, cells will have to be subjected to shear stress by applying a flow in

a cyclic-dependent manner to mimic the heartbeat. This would require engineering of a specific technical set-up. Two limitations of our protocol are the likely embryonic nature of engineered valvular cells and the 2D environment of cells in culture. Indeed, valvular cells differentiate early in the outflow tract and atrio-ventricular canal cushions filled with a jelly and mature up to weeks after birth in an extracellular matrix[3] and in a mechanically dependent manner[71]. These situations may in the future be mimicked by engineering hydrogels with appropriate stiffness and relevant components of the jelly and extracellular matrix proteins. HPVCs cells could then be plated on or embedded in 3D gels to better mimic the in vivo scenario.

This human model of valvulogenesis could additionally be applied to other patient-specific iPS cells and may help in understanding the molecular mechanisms underlying valvulo-pathies originating from defects in cell fate decisions and/or EMT.

## Methods

**Cell culture and sorting.** Human embryonic stem (HUES) cell lines HUES-24 and HUES-9, were obtained from Harvard Stem Cell Center (Dr Chad Cowan)[23]. HUES-24 was here used to generate a transgenic cell line expressing GFP under the control of the Sox9 promoter. To engineer the DNA construct, the 298-bp minimal *Sox9* promoter and two cis-regulatory elements[64] were excised from the pGBW101–1,2 vector (kindly given by Dr Bien-Willner, Baylor College of Medi-cine, Houston) and subcloned in a pAcEGFP1–1 vector. The DNA construct was electroporated and cells were selected with G418 for 10 days. HUES cell lines were cultured on Mouse Embryonic Fibroblasts (MEF) prepared from E14 Mouse embryos using KO-DMEM medium supplemented with β-mercaptoethanol ($10^{-7}$ M), glutamine, non-essential amino acids, 15% Serum Replacement (Thermo-fisher) and 10 ng/ml FGF2.

To generate a MesP1+ cell population[23], human pluripotent stem cells (both HUES and hiPS cells) cultured on MEFs were treated with first Wnt3a (100 ng/ml) and then Wnt3a (100 ng/ml) together with 10 ng/ml BMP2 for one day and then with 10 ng/ml BMP2 alone for one more day, in RPMI/B27 containing 1 μM SU5402, a FGF receptor inhibitor. For sorting, trypsinized cells were incubated for 30 min with gentle occasional agitation with EasySep™ Human Whole Blood CD15 (SSEA-1) Positive Selection Kit (Stem cells technologies) (25 μl/ml suspension cells) in D-PBS supplemented with 0.5% (wt/vol) Bovine Serum Albumin (BSA) and 2 mM EDTA at room temperature. Cells were then transferred to a column on a magnet. Cells were washed three times with 5 ml D-PBS-BSA/EDTA. Immunofluorescence, using an anti-CD15-FITC (Amicon) antibody, carried out directly after sorting revealed 90% purity of SSEA-1+ cells[23]. HUES or hiPS cells derived SSEA1+ sorted cells were phenotyped by RT-quantitative PCR and expressed MESP1, MEF2C, NKX2.5, ISL1 but not CD31, ENG or CDH5. Immunostaining documented that 90% of cells were MESP1+ [23].

To direct differentiation of (CD15+) SSEA-1+ cells towards endocardial HPVC fate, cells were cultured on MEFs plated at low density (10,000 cells/cm2) on fibronectin-coated plates and treated with 30 ng/ml VEGF, 10 ng/ml FGF8 and 2 ng/ml FGF2 for 6 days and sorted with anti-CD31 conjugated magnetic beads (Miltenyi, France) (Supplementary Fig 1). A step-by-step protocol describing in detail the valvular cell differentiation process has been described[65]. Separately, CD31+ cells cultured on fibronectin-coated plates were further induced to undergo EMT using 100 ng/ml BMP2 for 2 days in the presence or absence of DAPT (1 μM). In a series of experiments, 50 000 cells were transfected with a NCDI (Notch intracellular domain) expression plasmid (1 μg) or empty backbone vector using lipofectamine 2000.

**iPS cells.** iPS cells were derived using the Sendai viral vectors (Lifetech themofisher France). Cells were characterized by immunostaining for OCT4 (anti-OCT4 santa-cruz sc-9901), Sox2 (anti-SOX2 santa-cruz sc-17320) and NANOG (anti-NANOG R&D AF1997). Cells were then differentiated toward endoderm by treatment with 100 ng Activin in DMEM supplemented with 10% FCS for 3 days, or toward mesoderm by adding CHIR-99021 (5μM) the first day, CHIR-99021 and BMP2 (10 ng/ml) the second day and IWR1+ BMP2 the third day. Ectodermal cells were obtained by culturing cells for three days in RPMI + N2 supplement and 0.5 μM retinoic acid. Genomic integrity was tested at passages 15–20 using digital PCR of copy number variants of main human recurrent genomic abnormalities (stemgen-omics, Montpellier, France).

**Reverse-Transcription Real-time quantitative PCR.** RNA was extracted from SSEA1+ cells or VEGF/FGF8/FGF2-treated using a Zymo research kit. RNA (1μg) was reverse-transcribed using the Superscript II reverse transcriptase (Invitrogen, Cergy, France) and oligo(16)dT primers. Q-PCR was performed using SYBR Green and a Light Cycler LC 1.5 (Roche Diagnostic). Amplification was carried out as recommended by the manufacturer. Data analysis and primers specific for human genes are described in[23] and in Supplementary Data 2.

**Microarrays.** The AVC and the primary ventricles of E9.0 mouse embryos were dissected out and RNA (triplicates) was extracted with a Zymo research kit. RNA was also extracted from HPVCs. cRNAs were profiled using Illumina Mouse WG-6 v2 BeadChips or human Expression HG-U133 arrays and analyzed in Genespring GX 11.0. Data were quality filtered to exclude signal intensities below background, and expression profile differences of 4-fold or greater at a significance threshold of 0.01 or less were compiled into upregulated and downregulated transcript lists. Bioinformatically filtered genes were used for downstream analysis in Ingenuity Pathway software to examine network relationships as well as identify over-represented gene ontologies.

Cluster analysis of AVC and mesenchymal stem cell data sets were conducted as follows. Previously published AVC (GDS3663) and mesenchymal array data sets (GDS1288) were downloaded from GEO. HPVC gene expression profiles were examined using HG-U133 arrays and signal values were computed using GCRMA normalization. All expression signals for each dataset were Log10 transformed and normalized by z-score transformation[66]. Three experimental replicates were then averaged for each study, human and mouse orthologs and/or homologs between microarray data sets were determined by matching NCBI gene symbols across experiments and between species. This resulted in 9,385 orthologues that were then clustered using the HOPACH algorithm for gene clustering with the cosangle distance metric[67] or by hierarchical clustering with the Euclidean distance metric.

**Single cell-sequencing.** HiPS cell-derived HPVC and hiPS cell-derived valvular interstitial cells were dissociated with trypsin into single cells and processed with the SingleCell3 Reagent Kit on the Chromium platform as described by the manufacturer (10X genomics). cDNA libraries were sequenced with a Next-seq Illumina sequencer. A first analysis was performed with the Cell Ranger and loop cell10X genomics softwares. Clusters and subclusters were defined using genes that were differentially expressed in cell clusters in comparison with all other cells with a threshold of Log2 equal to at least 2. A secondary analysis was performed by B Jagla at the Pasteur Institute (Paris) using the SCDE package and scShinyHub (publicly available at https://github.com/baj12/scShinyHub). 1895 genes were detected as expressed in 50% of cells and 4576 in 25% of cells (Supplementary Fig. 2).

**Antibodies.** Antibodies used for cell or embryo immunofluorescence were used at a dilution of 1/100 unless otherwise specified. Antibodies were raised against Isl1 (Developmental Hybridoma bank, Iowa University 1/50), Tbx20 (novus biological H00057057-B01), Msx1 (Abcam ab93287), Tbx3 (Santa Cruz sc-178721), aggrecan (Chemicon AB1031), versican (Chemicon AB1033), smooth muscle actin (Sigma A-2547), Filamin A (Epitomics, CA, USA), vimentin dylight 550 (Novus biological, VM452, 1/200), periostin (Abcam ab140141), CD31 (BD pharmigen, WM59), VE-cadherin (R&D systems/MAB9381), Sox9 (a gift from Pr Wegner, University of Nurnberg, and Santa-Cruz Sc17431), GATA5 (Abcam ab11877), Hyaluronan-binding protein (Millipore 385911), collagen I (Abcam 34710), anti-Patched1 (Merck-Millipore, France, 06–1102), human Lamin A/C (Novacastra, NCL-LAM-A/C), and anti-NFATc (Santa-Cruz 17844 H10).

**Cell imaging.** High content imaging was performed in 96 wells plates using an Arrayscan (Cellomics Thermo Fisher Scientific) attached to an inverted microscope (Carl Zeiss, Inc.) using 20x N-Achroplan objective, NA 0.45, at room temperature. Other images were observed in epifluorescence microscopy (Zeiss microscope) or using an Ultraview Vox Spinning disk Perkin Elmer confocal microscope driven by the Volocity software or using a TRIO multispectral analysis setup (Caliper, Sci-ence). To visualize cilia and Ptch1 staining, stacks of images were acquired using a Zeiss observer epifluorescence microscope using a Plan-Apochromat 63X water objective or a confocal LSM800 Zeiss microscope equipped with an airyscan of 32 detectors. Light was provided by a Colibri 7 source or a laser module 405/488,561 and 640 nm wavelengths, respectively. Images were acquired using the ZEN ZEISS software. Then some images were deconvoluted using Autoquant and recon-structed in 3D using Imaris software (IMARIS). Quantification of extracellular matrix proteins was performed by thresholding images acquired with a Zeiss observer microscope or a confocal LSM800 equipped with a Colibri7 4LED light source using the same LED intensity and same exposure time for the CCD camera for all images or a laser module using a constant laser power. The surface of the cell field labelled by the antibody was calculated using Image J (NIH image). All details of image acquisition including excitation wavelengths and power of the light source, emission wavelengths and detector sensitivity as well image processing can be found in Supplementary Table 1, "Cell imaging parameters". All samples were mounted in Fluoromount™ (Cliniscience, France).

**AVC and OFT explants.** Dissection and culture of explants on collagen I hydrogel were carried out as described[36]. Briefly, after dissection, the OFT or AVC explants were cultured with the endocardium facing the gel in DMEM/ITS medium.

**Tendinous and chondrogenic differentiations of HPVCs.** Chondrogenic differ-entiation was performed according to established methods[33] HPVCs were isolated from MEF using collagenase IV (Life Technologies, France), transferred into 15 mL of polypropylene centrifuge tubes (500,000 cells/tube) and gently centrifuged. The

resulting pellets were statically cultured in DMEM high-glucose medium with glutamine, penicillin/streptomycin and chondrogenic supplements (1X insulin-transferrin-selenium, 1 μmol/L dexamethasone, 100 μmol/L ascorbic acid-2-phosphate), and 10 ng/mL TGF-β1. After 3 weeks, RNA was extracted from pellets and subjected to RT-Q-PCR. Cell pellets were fixed in 4% paraformaldehyde overnight, embedded in paraffin, and sectioned in 5 μm slices. Sections were stained with anti-collagen1a and –Sox9 antibodies.

**Collagen gel culture**. SSEA-1- HUES cells, SSEA-1+ HUES cells or SSEA1+ HUES cells treated with VEGF and FGF8 and FGF2 (HPVCs) were cultured alone and aggregated. Preparation of collagen gel (1 mg/ml type I collagen from rat tail tendon, BD Sciences) was described[68]. Resultant cell aggregates were placed on hydrated collagen gels and cultured for 48–72 h.

Cell injection in embryos, embryo culture, and isolated embryonic heart culture.

Mouse embryos were collected from E10.5 pregnant mice and cultured in DMEM or in M2 medium supplemented with 10% FCS at 37 °C and set with pins (isolated heart culture) or between pins (whole embryo culture) in a culture dish filled with silicone (Fig. 4a). Glass injection pipettes were pulled with a P-87 Flaming/Brown Micropipette Sutter Puller to get a 20 μm tip inner diameter. To optimize injection, a pipette was filled with the green dye CDCFDA, SE (Invitrogen/Molecular Probes, USA) and the dye injected into AVC.

Micropipettes were filled with HUESC-derived HVPC, or MesP1+ SSEA1+ or SSEA1- cells as controls in DMEM-10% FCS medium at a concentration of $10^5$cells/μl. One to three injections at a pressure of 200 hPa for 1 s were carried out in both the wall and the lumen of the AVC or OFT. Cell injections in brain were used as a negative control. At 4 h following injection, hearts were dissected and cultured at the air-liquid interface on insert coated with Matrigel and set in multiwell plates filled with DMEM-10% FCS. After 48 h culture, embryonic hearts were PFA-fixed and embedded in 1% agarose blocks prior to paraffin embedding. Alternatively, cells were injected in the AVC of E9.5 embryos through the yolk sac and embryos cultured in rolling tubes in 25% M16 medium /75% rat inactivated serum and gassed with 40% O2/5%CO2 for 24–48h[69].

**Statistics**. The data are expressed as mean ± SEM and represented as boxes and whiskers (min to max) showing the values lower than the 2.5th percentile and greater than the 97.5th percentile as circles. Experiments were repeated up to 9 times. Student-t test was used to compare data sets after checking for continuous probability distribution (Gaussian distribution) for each data group.

## Data availability

The authors declare that all data supporting the findings of this study are available within the article and its supplementary information files or from the corresponding author upon reasonable request. The raw data for the transcriptomic data have been deposited in the Genbank database under accession code: GSE73546. The source data underlying Figs. 1a, 2a, 2c and 5e are provided as a Source Data file.

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

## Acknowledgements

We are grateful to the UCSD Neuroscience Microscopy Shared Facility (supported by a grant P30 NS047101) for access to the Spinning disk microscope, to Jeff Smith and Dr Joan Heller Brown (UCSD, dept of Pharmacology, UCSD) for kindly allowing us to use their microinjection set up, and to Daniel Stockolm (Genethon, Evry, France) for access to the cell imaging facility. We are also thankful to Dr Valérie Balme and colleagues at HALIODX (Marseille) for helpful discussions and skilful single-cell-seq experiments. We thank the Leducq Fondation for supporting Tui Neri, and funding this research under the framework of the MITRAL network and for generously awarding us for the equipment of our cell imaging facility in the frame of their program "Equipement de Recherche et Plateformes Technologiques" (ERPT to M.P.), the Genopole at Evry and the Fondation de la recherche Medicale (grant DEQ20100318280) for supporting the laboratory of Michel Puceat. Part of this work in South Carolina University was conducted in a facility constructed with support from the National Institutes of Health, Grant Number C06 RR018823 from the Extramural Research Facilities Program of the National Center for Research Resources. Other funding sources: National Heart Lung and Blood Institute: RO1-HL33756 (R.R.M.), COBRE P20RR016434–07 (R.R.M., R.A. N.), P20RR016434–09S1 (R.R.M. and R.A.N.); American Heart Association: 11SDG5270006 (R.A.N.); National Science Foundation: EPS-0902795 (R.R.M. and R.A. N.); American Heart Association: 10SDG2630130 (A.C.Z.), NIH: P01HD032573 (A.C. Z.), NIH: U54 HL108460 (A.C.Z), NCATS: UL1TR000100 (A.C.Z); EH was supported by a fellowship of the Ministere de la recherche et de l'éducation in France.TM-M was supported by a fellowship from the Fondation Foulon Delalande and the Leducq Foundation. P.v.V. was sponsored by a UC San Diego Cardiovascular Scholarship Award and a Postdoctoral Fellowship from the California Institute for Regenerative Medicine (CIRM) Interdisciplinary Stem Cell Training Program II. S.M.E. was funded by a grant from the National Heart, Lung, and Blood Institute (HL-117649). A.T. is supported by the National Heart, Lung, and Blood Institute (R01-HL134664).

## Author contributions

T.N. and E.H. designed and performed in vitro and ex vivo experiments. P.V.V. designed and performed in vivo experiments with mouse embryos and designed Supplementary Figure 1. EF performed ISH experiments. R.A.N. and YS performed ex vivo and in vivo experiments, respectively. BF derived hiPS cells. B.J. implemented scShinyHub and helped with bioinformatics for single cell experiments. J.L. performed experiments with *Dachsous* mutated hiPS cells. T.M.M. helped with ex vivo experiments. S.Z. collected mouse embryos and dissected AVC for transcriptomic analysis. R.F.S. and AZC analyzed microarray data. J.P.D. and D.S. performed analysis of single cell data. R.A.N., R.L., J.L.P., A.T., R.M. edited the manuscript. M.P. designed and coordinated the study, performed experiments, and wrote the manuscript.

## Additional information

**Competing interests:** The authors declare no competing interests.

