## [Peer Review File · Nature Communications]

Reviewers' Comments:

Reviewer #1:

Remarks to the Author:

In this article, Neri/Hiriart et al. are describing a novel protocol to derive cardiac valve progenitors from human pluripotent stem cells. The authors propose that such cultured cells can help to study the early steps of human cardiac valve development. I found that the article is well written and that it provides a solid analysis of the generated cells along with a nice demonstration of the application of such in vitro model to study human heart valve disease. Overall I am in favor for publication.

There are just a few points that could be addressed by the authors :

-While the authors make a great introduction/discussion about the need of having the possibility human valve in a dish, they do not really discuss the weaknesses of their model. I found that the 'future consideration' part is a great addition to the discussion, and I feel that the community would benefit from the study if the authors could discuss more clearly the drawbacks of this cellular model and how to bypass them.

-An intriguing part of the study is that the progenitors do not need to be stimulated by mechanical forces to properly express several valve identity and EMT markers. The authors should discuss their results in light of the recent findings describing the endocardial cell response to flow forces. What about *Klf2/4* level of expression in these cells ? In addition Wnt signaling seems important to relay the endocardial cell response to flow into mesenchymal cells. How are these observations fitting with the generated in vitro model? This is an important point because flow forces and endogenous mechanical stimuli seem to be a major contributor to valve identity.

-for clarity, the authors should provide a drawing explaining the different steps of the differentiation procedure including parameters of length/purity/molecules used etc.

Reviewer #2:

Remarks to the Author:

In this manuscript, the authors describe a novel concept of generating an in vitro method of producing a unique population of cells from human pluripotent cells which are suited for studying , in vitro, , mechanisms of valvulogenesis or inherited valve disease.. the pluripotent cells were first differentiated into "human prevalvular cells HPVC" which were manipulated further to produce more mature valve like tissue " valve in a dish" they then experiments both in vitro and in vivo in mice, to demonstrate the utility of their in vitro model in studying early human valvulogenesis. In addition they examined the potential of the model in studying the molecular mechanisms of mitral valve prolapse in valve interstitial cells obtained from an excised mitral valve from patient carrying a mutation in a gene recently identified to be a cause of MVP. Further studies are required to validate their early results.

Reviewer #3:

Remarks to the Author:

The manuscript entitled „A human cell model of cardiac pathophysiological valvogenesis" by Neri et al. describes for the first time the generation of a specialized valvular cell type from human pluripotent stem cells (ESC and iPSC). The authors termed these cells "human pre-vascular cells" (HPVCs) and demonstrated their resemblance to primary cells derived from the murine atrio-ventricular canal (AVC) via extensive transcriptomic phenotyping including single cell PCR analysis. HPVCs were shown to undergo EMT in a Notch dependent fashion in vitro, ex vivo and in vivo in a

mouse embryo model. Patient specific iPSCs harboring a Dachshous mutation were generated and used to model some aspects of valve disease (namely mitral valve prolapse) in a dish. These cells could be used to generate Dachshous mutated HPVCs, which displayed impaired cilia formation, excessive secretion of hyaluronan, collagen I and periostin, and showed aberrant specification after EMT induction. The authors could show a role for sonic hedgehog (SSH) signaling in these processes and certain aspects of the disease phenotype were rescued by addition of an SHH inhibitor, thus revealing a potential target for treatment.

These findings are novel and original and will be of interest to others in the field of stem cell research and cardiac development. The design of the study is straightforward and the conclusions seem well justified from the data presented.

The methods are described briefly but in sufficient detail, including media composition and concentrations of the growth factors and chemicals used. However, the timing of treatment steps is not entirely clear and varies throughout the manuscript:

- According to the Materials and Methods section, the differentiation of pluripotent stem cells was induced with 100 ng/ml Wnt3a, 10 ng/ml BMP2 and 1 μ M SU5402 for 4 days (line 155). In contrast, the results section states: "Undifferentiated pluripotent stem cells were first differentiated into MesP1+ cardiovascular progenitors using Wnt3a (100 ng/ml) and BMP2 (10 ng/ml) for two days and then BMP2 alone" (line 293). Please clarify.
- According to the Materials and Methods section, SSEA1+ cells were treated with 30 ng/ml VEGF, 10 ng/ml FGF8 and 2 ng/ml FGF2 for 6 days (line 168). However, only VEGF and FGF8 are mentioned in the figure legends 1-4, with a treatment duration of 6 days, 1 week and 7 days (lines 883, 902, 924, and 942). Please clarify.
- It would be very helpful for the reader to include a schematic representation of the protocol for differentiation (as in Protze et al. Nature Biotechnology 2017). This should include a clear labeling of the intermediate cell populations, e.g. to make clear when exactly HPVCs are generated. Labeling of the cell populations is somewhat confusing in the text: In Materials and Methods, sorting is described only for SSEA1 (line 158) and CD31 (line 169), but in the Results, "the MESP1+-sorted cell populations" are mentioned (line 304). Please clarify.

Additional minor comments:

Lines 325-348: description of clusters 4 & 6 is missing?

Lines 389-391: This sentence is just rephrasing the previous one (line 385-387). Please check.

Line 459: "insets Figure 5e" should read "insets Figure 5d".

Line 460: "Fig.5f" should read "Fig. 5e".

Line 521: "The ensuing application of patient specific iPSC cell-derived valvular cells to model replay valvulopathy exemplifies a genuine model of patient-specific nonsyndromic mitral valve prolapse underscoring the biological validity and clinical utility of this proof-of-concept study." Wording? "model" or "replay"?

Line 580: "We have found that FGF8, known to favor endocardium vs myocardium as well as in the formation and EMT of cardiac cushion, was [...] " Please check sentence for missing word(s)?

Line 629: "Interestingly, cilia signals through sonichedghog [...]" should read "Interestingly, cilia signal through sonic hedgehog [...]"?

Figure 3c: one label "huLMNA" is missing

Alignment of figure panels: Figure 1d; Figure 2d; Figure 4 a, b, d; Figure 5d, f; Figure 6b.

Scale bars in Figure 4a are missing.

Reply to Reviewers

We would like first to thank the referees for the positive reviews of our study. We did appreciate their helpful comments. We thus fully followed their advices and have corrected our manuscript accordingly.

Reviewer #1 (Remarks to the Author):

In this article, Neri/Hiriart et al. are describing a novel protocol to derive cardiac valve progenitors from human pluripotent stem cells. The authors propose that such cultured cells can help to study the early steps of human cardiac valve development. I found that the article is well written and that it provides a solid analysis of the generated cells along with a nice demonstration of the application of such in vitro model to study human heart valve disease. Overall I am in favor for publication.

There are just a few points that could be adressed by the authors :

-While the authors make a great introduction/discussion about the need of having the possibility human valve in a dish, they do not really discuss the weaknesses of their model. I found that the 'future consideration' part is a great addition to the discussion, and I feel that the community would benefit from the study if the authors could discuss more clearly the drawbacks of this cellular model and how to bypass them.

Thank you for that useful comment. We now further discussed p15 in the paragraph **Future considerations** the limitations of the current cell model and how it could be improved.

P15:

Our in vitro, ex vivo and in vivo assays facilitate further studies examining the events that give rise to the human endocardium, valvular interstitial cells, and tendinous/chondrogenic cells, as well as to delineate the pathways triggering EMT and potentially underlying mechanotransduction. For such an aim, cells will have to be subjected to shear stress by applying a flow in a cyclic-dependent manner to mimic heartbeat. This would require engineering of a specific technical set-up. Two limitations of our protocol are the likely embryonic nature of engineered valvular cells and the 2D environment of cells in culture. Indeed, valvular cells differentiate early in the outflow tract and atrioventricular canal cushions filled with a jelly and mature up to weeks after birth in an extracellular matrix³ and in a mechanical dependent manner⁷¹. These situations may in the future be mimicked by engineering hydrogels with a correct stiffness with components of the jelly and then with extracellular matrix proteins. Our cells could then be plated on or embedded in 3D in such gels to better mimic the in vivo scenario.

This human model of valvulogenesis could be extended to patient-specific iPS cells and will help in understanding the molecular mechanisms underlying valvulopathies originating from defects in cell fate decisions and/or EMT.

-An intriguing part of the study is that the progenitors do not need to be stimulated by mechanical forces to properly express several valve identity and EMT markers. The authors should discuss their results in light of the recent findings describing the endocardial cell response to flow forces. What about klf2/4 level of expression in these cells ? In addition Wnt signaling seems important to relay the endocardial cell response to flow into mesenchymal cells. How are these observations fitting with the generated in vitro model? This is an important point because flow forces and endogenous mechanical stimuli seem to be a major contributor to valve identity.

Thank you for this very exciting comment. We indeed are aware of the role of the hemodynamic flow in patterning the valves from early steps of differentiation to maturation. We looked in our single cells data. *KLF2* is expressed in a subset of endothelial pre-EMT cells (161 cells out of 729 cells), now added p8 last paragraph. We do not have unfortunately available engineered set up to impose a hemodynamic flow to our cells. We however tested the Wnt response of HPVCs. Wnt3a was chosen as an agonist as this was the only Wnt member expressed in our single cell data. As shown in supplementary figure 5 (described p10), *KLF2* was significantly increased upon Wnt stimulation, which suggests from the literature (reference 71) a positive feedback response of Wnt, *klf2* inducing *wnt9a* and further responding to wnt signaling. We agree that BMP2, Wnt and notch are all key transduction pathway in valvulogenesis. We have thus also tested whether Wnt induced EMT genes in our cell model. Indeed, as illustrated in supplementary Figure 5c, Wnt stimulation strongly upregulated *SLUG*, *MSX1* and *SMAD6*, index of ongoing EMT. These data are now described p10 (first paragraph).

-for clarity, the authors should provide a drawing explaining the different steps of the differentiation procedure including parameters of length/purity/molecules used etc.

We have added a cartoon that illustrates step-by-step our protocol of differentiation (supplementary figure 1) and have written and posted the full protocol (including the references of molecules used), in detail to protocol exchange Nature web site.

-

Reviewer #2 (Remarks to the Author):

In this manuscript, the authors describe a novel concept of generating an in vitro method of producing a unique population of cells from human pluripotent cells which are suited for studying , in vitro, , mechanisms of valvulogenesis or inherited valve disease.. the pluripotent cells were first differentiated into “human preavalvular cells HPVC” which were manipulated further to produce more mature valve like tissue “ valve in a dish” they then experiments both in vitro and in vivo in mice, to demonstrate the utility of their in vitro model in studying early human valvulogenesis. In addition they examined the potential of the model in studying the molecular mechanisms of mitral valve prolapse in valve interstitial cells obtained from an excised mitral valve from patient carrying a mutation in a gene recently identified to be a cause of MVP. Further studies are required to validate their early results.

We would like to thank the referee who acknowledged the novel concept brought by our study.

Our manuscript includes 7 figures and 8 supplementary figures. We report a detailed and step-by-step protocol of differentiation of pluripotent stem cells into valvular cells. We further studied the potential and behavior of cells *in vitro*, *ex-vivo* and *in vivo*. Finally, using a patient-specific iPSC model of mitral valve prolapse, we uncovered a signal transduction pathway whose dysregulation is at least in part causative of the disease. We do believe in the strength and in the scientific (in a developmental biology point of view) and clinical relevance of our study. We are sorry; we indeed could not understand what the referee was expecting from our documented research.

Reviewer #3 (Remarks to the Author):

The manuscript entitled "A human cell model of cardiac pathophysiological valvogenesis" by Neri et al. describes for the first time the generation of a specialized valvular cell type from human pluripotent stem cells (ESC and iPSC). The authors termed these cells "human pre-vascular cells" (HPVCs) and demonstrated their resemblance to primary cells derived from the murine atrio-ventricular canal (AVC) via extensive transcriptomic phenotyping including single cell PCR analysis. HPVCs were shown to undergo EMT in a Notch dependent fashion in vitro, ex vivo and in vivo in a mouse embryo model. Patient specific iPSCs harboring a Dachshous mutation were generated and used to model some aspects of valve disease (namely mitral valve prolapse) in a dish. These cells could be used to generate Dachshous mutated HPVCs, which displayed impaired cilia formation, excessive secretion of hyaluronan, collagen I and periostin, and showed aberrant specification after EMT induction. The authors could show a role for sonic hedgehog (SSH) signaling in these processes and certain aspects of the disease phenotype were rescued by addition of an SHH inhibitor, thus revealing a potential target for treatment.

These findings are novel and original and will be of interest to others in the field of stem cell research and cardiac development. The design of the study is straightforward and the conclusions seem well justified from the data presented.

The methods are described briefly but in sufficient detail, including media composition and concentrations of the growth factors and chemicals used. However, the timing of treatment steps is not entirely clear and varies throughout the manuscript:

- According to the Materials and Methods section, the differentiation of pluripotent stem cells was induced with 100 ng/ml Wnt3a, 10 ng/ml BMP2 and 1 μ M SU5402 for 4 days (line 155). In contrast, the results section states: "Undifferentiated pluripotent stem cells were first differentiated into MesP1+ cardiovascular progenitors using Wnt3a (100 ng/ml) and BMP2 (10 ng/ml) for two days and then BMP2 alone" (line 293). Please clarify.

- According to the Materials and Methods section, SSEA1+ cells were treated with 30 ng/ml VEGF, 10 ng/ml FGF8 and 2 ng/ml FGF2 for 6 days (line 168). However, only VEGF and FGF8 are mentioned in the figure legends 1-4, with a treatment duration of 6 days, 1 week and 7 days (lines 883, 902, 924, and 942). Please clarify.

We thank the referee for acknowledging the novelty and originality of our work. We are sorry for the inconsistencies in the methods and results sections about details of the protocol of cell differentiation. We have now corrected these mistakes in the manuscript and have written a detailed protocol posted in Nature protocol exchange that we hope clarifies this point.

- It would be very helpful for the reader to include a schematic representation of the protocol for differentiation (as in Protze et al. Nature Biotechnology 2017). This should include a clear labeling of the intermediate cell populations, e.g. to make clear when exactly HPVCs are generated. Labeling of the cell populations is somewhat confusing in the text: In Materials and Methods, sorting is described only for SSEA1 (line 158) and CD31 (line 169), but in the Results, "the MESP1+-sorted cell populations" are mentioned (line 304). Please clarify.

We are sorry for this confusion. We have reported (Blin et al., JCI 2010) that BMP2-induced SSEA1 and in turn SSEA1⁺ sorted cells are MESP1⁺. We have however clarified this point in the MS by naming the cells SSEA1⁺, MESP1⁺. We have added a cartoon of the detailed protocol (Supplementary figure 1) including the labeling of cell populations

Additional minor comments:

Lines 325-348: description of clusters 4 & 6 is missing? Thank you; Now described

Lines 389-391: This sentence is just rephrasing the previous one (line 385-387). Please check. Thank you; now corrected

Line 459: "insets Figure 5e" should read "insets Figure 5d". Thank you; now corrected

Line 460: "Fig.5f" should read "Fig. 5e". Thank you; now corrected

Line 521: “The ensuing application of patient specific iPS cell-derived valvular cells to model replay valvulopathy exemplifies a genuine model of patient-specific nonsyndromic mitral valve prolapse underscoring the biological validity and clinical utility of this proof-of- concept study.” Wording? “model” or “replay”?

Thank you; now corrected

Line 580: “We have found that FGF8, known to favor endocardium vs myocardium as well as in the formation and EMT of cardiac cushion, was [...] ” Please check sentence for missing word(s)? Thank you; now corrected

Line 629: “Interestingly, cilia signals through sonichedghog [...]” should read “Interestingly, cilia signal through sonic hedgehog [...]”? Thank you; now corrected

Figure 3c: one label “huLMNA” is missing Thank you; now corrected

Alignment of figure panels: Figure 1d; Figure 2d; Figure 4 a, b, d; Figure 5d, f; Figure 6b.

Scale bars in Figure 4a are missing. Thank you; now aligned

Reviewers' Comments:

Reviewer #3:

Remarks to the Author:

I was happy to see that in their revised version of the manuscript entitled „A humal cell model of cardiac pathophysiological valvogenesis“ the authors were able to provide clarification for a number of points. As said before, their findings are novel and original and will be of interest to others in the field of stem cell research and cardiac development. Therefore, I can only recommend the publication of the manuscript in its current version.